# On the Derivation of Winograd-Type DFT Algorithms for Input Sequences Whose Length Is a Power of Two

**Mateusz Raciborski** *,† and **Aleksandr Cariow** †

Faculty of Computer Science and Information Technology, West Pomeranian University of Technology, Żołnierska 52, 71-210 Szczecin, Poland; atariov@wi.zut.edu.pl
* Correspondence: mateusz.raciborski@zut.edu.pl
† These authors contributed equally to this work.

**Abstract:** Winograd's algorithms are an effective tool for calculating the discrete Fourier transform (DFT). These algorithms described in well-known articles are traditionally represented either with the help of sets of recurrent relations or with the help of products of sparse matrices obtained on the basis of various methods of the DFT matrix factorization. Unfortunately, in the mentioned papers, it is not shown how the described relations were obtained or how the presented factorizations were found. In this paper, we use a simple, understandable and fairly unified approach to the derivation of the Winograd-type DFT algorithms for the cases $N = 8$, $N = 16$ and $N = 32$. It is easy to verify that algorithms for other lengths of sequences that are powers of two can be synthesized in a similar way.

**Keywords:** complexity theory; compression algorithms; digital signal processing; discrete Fourier transforms; fast Fourier transforms; matrix decomposition; signal processing algorithms; sparse matrices; sum product algorithm; Winograd discrete Fourier transform algorithm





## 1. Introduction

Winograd's method for the realization of the discrete Fourier transform (DFT) for several decades has been discussed in a number of publications [1–12]. In comparison with the Cooley–Tukey fast Fourier transform (FFT) algorithms, the Winograd DFT algorithm requires substantially fewer multiplications at the cost of a few extra additions. In the known papers, the cases of the Winograd FFTs for small sequences of odd length are mainly considered. Moreover, the algorithms were presented in the form of algebraic relations or in the form of DFT matrix factorizations. However, none of the publications known to us has written on how these relations were obtained or how, on the basis of any considerations, the matrices that make up the corresponding computational procedures were constructed.

In this paper, we want to show a simple, understandable and fairly unified approach to the derivation of Winograd-type FFT algorithms for the cases $N = 8$, $N = 16$ and $N = 32$. It is easy to verify that algorithms for other lengths of sequences that are powers of two can be synthesized similarly.

## 2. Preliminary Remarks

The discrete Fourier transform (DFT) is one of the most important tools in digital signal and image processing. The DFT can be defined as follows:

$$y_k = \sum_{n=0}^{N-1} x_n e^{\frac{-j2\pi nk}{N}} \tag{1}$$

where $x_n, n = 0, 1, \ldots, N - 1$ is a uniformly sampled sequence, $y_k, k = 0, 1, \ldots, N - 1$ is the $k$-th DFT coefficient, and $j = \sqrt{-1}$ is an imaginary unit.

In vector–matrix notation, we can rewrite (1) in the following form:

$$\mathbf{Y}_{N\times 1} = \mathbf{E}_N \mathbf{X}_{N\times 1} \tag{2}$$

where

$$\mathbf{X}_{N\times 1} = [x_0, x_1, \ldots, x_{N-1}]^\mathrm{T}, \mathbf{Y}_{N\times 1} = [y_0, y_1, \ldots, y_{N-1}]^\mathrm{T}$$

and

$$\mathbf{E}_N = \| w^{kn} \|, w^{kn} = e^{\frac{-j2\pi nk}{N}}, k, n = 0, 1, \ldots, N-1$$

Implementation of calculations in accordance with expression (2), especially for large $N$, requires performing a large number of arithmetic operations, which in turn leads to an increase in computation time.

In 1965, J. Cooley and J. Tukey proposed the fast algorithm to compute discrete Fourier transform with a drastically reduced number of arithmetical operations. Mathematically, the fast Fourier transform algorithms are based on factorization of the Fourier matrix into a product of sparse matrices, meaning matrices with many zero entries. However, this factorization can be implemented in different ways. In the case of the Cooley–Tukey algorithm, we are dealing with the representation of the original matrix as a product of $\log_2 N$ sparse structured matrices. As is well known, the complexity of this algorithm is approximately $\frac{N}{2} \log_2 N$ multiplications and the same number of additions of complex numbers.

Another effective algorithm for calculating the DFT is the Winograd FFT algorithm. In comparison with the Cooley–Tukey FFT algorithm, the Winograd FFT algorithm requires substantially fewer multiplications at the cost of a few extra additions. Winograd proved that the multiplicative complexity of FFT algorithms can be significantly reduced by some increase in additive complexity. The Winograd Fourier transform algorithm (WFTA) is an FFT algorithm which achieves a reduction on the number of multiplications from order $\mathcal{O}(N2)$ in the DFT to order $N$. In the literature known to the authors [1–15], the Winograd FFT algorithms that implement DFT transform for a limited set of small-length sequences are mainly considered. As a rule, these algorithms are represented as a set of algebraic relations [1–5,8], although the matrix interpretation of Winograd FFT algorithms are available too [6,9]. In the case of the matrix formulation of the Winograd FFT algorithms, the factorization of the DFT matrix differs from the factorization of the DFT matrix in the Cooley–Tukey FFT algorithms. Moreover, the mechanism for deriving such algorithms for each specific case is unique. In addition, methods for the deriving of recurrent relations are not published anywhere. Additionally, the ways for deriving factorized representations of DFT matrices have never been explained. In this paper, we show a simple, understandable and fairly unified approach to the derivation of the Winograd-like FFT algorithms for the case when the input sequence length is a power of two.

## 3. Short Background

The main idea of the proposed approach is to use a new method for factorizing the DFT matrix, which is different from Winograd factorization. In contrast to Winograd factorization, we propose the following unified method of DFT matrix decomposition:

$$\mathbf{E}_{2^{i+1}} = (\mathbf{H}_2 \otimes \mathbf{I}_{2^i})(\mathbf{E}_{2^i} \oplus \mathbf{Q}_{2^i})\mathbf{P}_{2^{i+1}}^{(\pi_{2^{i+1}})} \tag{3}$$

where

$$\mathbf{P}_{2^{i+1}}^{(\pi_{2^{i+1}})} = \left[ \frac{\mathbf{I}_{2^{i-1}} \otimes \mathbf{\Psi}_{2\times 4}}{(\mathbf{I}_{2^{i-1}} \otimes \mathbf{\Psi}_{2\times 4})\mathbf{I}_{2^{i+1}}^{(1\rightarrow)}} \right], \mathbf{\Psi}_{2\times 4} = \left[ \begin{array}{cccc} 1 & 0 & 0 & 0 \\ 0 & 0 & 1 & 0 \end{array} \right], i = 1, 2, 3 \ldots$$

$\mathbf{E}_k$ is $k \times k$ DFT matrix; $\mathbf{Q}_k$ is some "prefix" matrix containing a constellation of twiddle factors specific for each $N$; $\mathbf{I}_k$ is an identity $k \times k$ matrix; $\mathbf{H}_2$ is the order 2 Hadamard matrix; $\mathbf{I}_{2^{i+1}}^{(1\rightarrow)}$ is the matrix obtained from the $k \times k$ identity matrix by shifting its columns by one

position to the right; and signs "$\otimes$", "$\oplus$" denote the tensor product and direct sum of two matrices, respectively [16,17].

Then, the generalized scheme for the synthesis of Winograd-type DFT algorithms for $N$ equal to the power of two can be described as follows:

$$\mathbf{Y}_{2^{i+1}\times 1} = (\mathbf{H}_2 \otimes \mathbf{I}_{2^i})(\mathbf{E}_{2^i} \oplus \mathbf{Q}_{2^i})\mathbf{P}_{2^{i+1}}^{(\pi_{2^{i+1}})}\mathbf{X}_{2^{i+1}\times 1} \tag{4}$$

The methods for factorizing the matrices $\mathbf{E}_k$ and $\mathbf{Q}_k$ are different, but both lead to a factorization of the **BCD** type [18] similar to the Winograd factorization. Moreover, as follows from expression (1), the expansions for small $N$ are part of the expansions for larger lengths of input sequences. When synthesizing algorithms for separate $\mathbf{E}_k$ and $\mathbf{Q}_k$, we will use the templates of matrix structures and identities presented in [19,20].

## 4. Synthesis of the Fast Winograd-Type DFT Algorithms

Let us show, based on specific examples, how it works.

### 4.1. Fast DFT Algorithm for N = 4

As an example, suppose that $N = 4$. Then (2) can be rewritten as

$$\mathbf{Y}_{4\times 1} = \mathbf{E}_4\mathbf{X}_{4\times 1} \tag{5}$$

where

$$\mathbf{E}_4 = \left[\begin{array}{cc:cc} a_4 & a_4 & a_4 & a_4 \\ a_4 & b_4 & -a_4 & -b_4 \\ \hdashline a_4 & -a_4 & a_4 & -a_4 \\ a_4 & -b_4 & -a_4 & b_4 \end{array}\right],$$

$$a_4 = 1, b_4 = -j.$$

$$\mathbf{X}_{4\times 1} = [x_0, x_1, x_2, x_3]^{\mathrm{T}}, \mathbf{Y}_{4\times 1} = [y_0, y_1, y_2, y_3]^{\mathrm{T}}.$$

Let us now define the permutation $\pi_4$ and write it as a matrix in this way:

$$\pi_4 = \begin{pmatrix} 1 & 2 & 3 & 4 \\ 1 & 3 & 2 & 4 \end{pmatrix}, \quad \mathbf{P}_4^{(\pi_4)} = \left[\begin{array}{cc:cc} 1 & & & \\ & & 1 & \\ \hdashline & 1 & & \\ & & & 1 \end{array}\right].$$

Permute the columns of the matrix $\mathbf{E}_4$ according to permutation $\pi_4$. As a result of such a permutation, we obtain the matrix

$$\tilde{\mathbf{E}}_4 = \left[\begin{array}{c:c} \mathbf{E}_2 & \mathbf{Q}_2 \\ \hdashline \mathbf{E}_2 & -\mathbf{Q}_2 \end{array}\right] = \tilde{\mathbf{E}}_4 \mathbf{P}_4^{(\pi_4)}$$

where

$$\mathbf{E}_2 = \left[\begin{array}{c:c} a_4 & a_4 \\ \hdashline a_4 & -a_4 \end{array}\right] \quad \text{and} \quad \mathbf{Q}_2 = \left[\begin{array}{c:c} a_4 & a_4 \\ \hdashline b_4 & -b_4 \end{array}\right]$$

According to the concept, Expression (4) for a given transform size can be rewritten as follows:

$$\mathbf{Y}_{4\times 1} = (\mathbf{H}_2 \otimes \mathbf{I}_2)(\mathbf{E}_2 \oplus \mathbf{Q}_2)\mathbf{P}_4^{(\pi_4)}\mathbf{X}_{4\times 1} \tag{6}$$

where

$$\mathbf{H}_2 \otimes \mathbf{I}_2 = \left[\begin{array}{c:c} 1 & 1 \\ \hdashline 1 & -1 \end{array}\right] \otimes \left[\begin{array}{c:c} 1 & 0 \\ \hdashline 0 & 1 \end{array}\right] = \left[\begin{array}{cc:cc} 1 & & 1 & \\ & 1 & & 1 \\ \hdashline 1 & & -1 & \\ & 1 & & -1 \end{array}\right] = \mathbf{W}_4^{(0)}.$$

As you can see, after rearranging the columns of the DFT matrix $\mathbf{E}_4$, it can be decomposed, as follows from the proposed technique, into the order 2 DFT matrix $\mathbf{E}_2$ and the order 2 prefix matrix $\mathbf{Q}_2$.

For matrices $\mathbf{E}_2$ and $\mathbf{Q}_2$, we can offer the following factorization schemes leading to a reduction in computational complexity:

$$\mathbf{E}_2 = \left[\begin{array}{c|c} a_4 & a_4 \\ \hline a_4 & -a_4 \end{array}\right] = (a_4 \oplus a_4)\mathbf{H}_2, \quad \mathbf{Q}_2 = \left[\begin{array}{c|c} a_4 & a_4 \\ \hline b_4 & -b_4 \end{array}\right] = (a_4 \oplus b_4)\mathbf{H}_2$$

Taking into account the above factorization schemes, we can finally write

$$\mathbf{Y}_{4\times1} = \mathbf{W}_4^{(0)}\mathbf{D}_4\mathbf{W}_4^{(1)}\mathbf{P}_4^{(\pi_4)}\mathbf{X}_{4\times1} \tag{7}$$

where

$$\mathbf{W}_4^{(1)} = \left[\begin{array}{c|c} 1 & 1 \\ \hline 1 & -1 \end{array}\right] \oplus \left[\begin{array}{c|c} 1 & 1 \\ \hline 1 & -1 \end{array}\right] = \left[\begin{array}{cc|cc} 1 & 1 & & \\ 1 & -1 & & \\ \hline & & 1 & 1 \\ & & 1 & -1 \end{array}\right],$$

$$\mathbf{D}_4 = diag(\varphi_0, \varphi_1, \varphi_2, \varphi_3),$$

$$\varphi_0, \varphi_1, \varphi_2 = a_4 = 1, \varphi_3 = b_4 = -j.$$

Figure 1 shows a data flow graph of synthesized algorithm for 4 point DFT. As can be seen, in this case, the algorithm takes only eight additions.

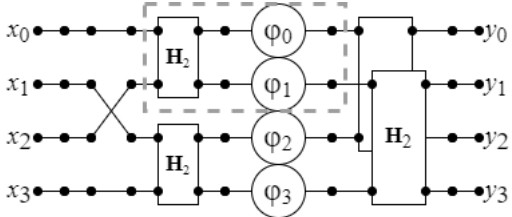

**Figure 1.** The data flow graph of the proposed algorithm for computation of 4-point DFT.

*4.2. Fast DFT Algorithm for N = 8*

As an example, suppose that $N = 8$. Then (2) can be rewritten as

$$\mathbf{Y}_{8\times1} = \mathbf{E}_8\mathbf{X}_{8\times1} \tag{8}$$

where

$$\mathbf{X}_{8\times1} = [x_0, x_1, x_2, x_3, x_4, x_5, x_6, x_7]^{\mathrm{T}},$$

$$\mathbf{Y}_{8\times1} = [y_0, y_1, y_2, y_3, y_4, y_5, y_6, y_7]^{\mathrm{T}},$$

$$a_8 = 1, b_8 = 0.7071 - j0.7071, c_8 = -j, d_8 = -0.7071 - j0.7071,$$

$$\mathbf{E}_8 = \left[\begin{array}{cccc|cccc} a_8 & a_8 & a_8 & a_8 & a_8 & a_8 & a_8 & a_8 \\ a_8 & b_8 & c_8 & d_8 & -a_8 & -b_8 & -c_8 & -d_8 \\ a_8 & c_8 & -a_8 & -c_8 & a_8 & c_8 & -a_8 & -c_8 \\ a_8 & d_8 & -c_8 & b_8 & -a_8 & -d_8 & c_8 & -b_8 \\ \hline a_8 & -a_8 & a_8 & -a_8 & a_8 & -a_8 & a_8 & -a_8 \\ a_8 & -b_8 & c_8 & -d_8 & -a_8 & b_8 & -c_8 & d_8 \\ a_8 & -c_8 & -a_8 & c_8 & a_8 & -c_8 & -a_8 & c_8 \\ a_8 & -d_8 & -c_8 & -b_8 & -a_8 & d_8 & c_8 & b_8 \end{array}\right].$$

Let us now define the permutation $\pi_8$ in the following form:

$$\pi_8 = \begin{pmatrix} 1 & 2 & 3 & 4 & 5 & 6 & 7 & 8 \\ 1 & 3 & 5 & 7 & 2 & 4 & 6 & 8 \end{pmatrix}.$$

Permutation $\pi_8$ can be written as a matrix in this way:

$$\mathbf{P}_8^{(\pi_8)} = \left[ \begin{array}{cccc:cccc} 1 & & & & & & & \\ & 1 & & & & & & \\ & & & & 1 & & & \\ & & & & & & & 1 \\ \hdashline & & 1 & & & & & \\ & & & 1 & & & & \\ & & & & & 1 & & \\ & & & & & & 1 & \end{array} \right].$$

Permute columns of the matrix $\mathbf{E}_8$ according to permutation $\pi_8$. As a result of such a permutation, we obtain the matrix

$$\tilde{\mathbf{E}}_8 = \left[ \begin{array}{c:c} \mathbf{E}_4 & \mathbf{Q}_4 \\ \hdashline \mathbf{E}_4 & -\mathbf{Q}_4 \end{array} \right] = \mathbf{E}_8 \mathbf{P}_8^{(\pi_8)}$$

where

$$\mathbf{E}_4 = \begin{bmatrix} a_8 & a_8 & a_8 & a_8 \\ a_8 & c_8 & -a_8 & -c_8 \\ a_8 & -a_8 & a_8 & -a_8 \\ a_8 & -c_8 & -a_8 & c_8 \end{bmatrix} \quad \text{and} \quad \mathbf{Q}_4 = \begin{bmatrix} a_8 & a_8 & a_8 & a_8 \\ b_8 & d_8 & -b_8 & -d_8 \\ c_8 & -c_8 & c_8 & -c_8 \\ d_8 & b_8 & -d_8 & -b_8 \end{bmatrix}$$

According to the concept, Expression (4) for a given transform size can be rewritten as follows:

$$\mathbf{Y}_{8\times 1} = (\mathbf{H}_2 \otimes \mathbf{I}_4)(\mathbf{E}_4 \oplus \mathbf{Q}_4)\mathbf{P}_8^{(\pi_8)}\mathbf{X}_{8\times 1} \tag{9}$$

Such a structure of the matrix $\tilde{\mathbf{E}}_8$ allows to apply a divide and conquer algorithm that recursively breaks down a matrix–vector product of order eight into two smaller matrix–vector products of order 4 [19]. If we write the matrix $\mathbf{E}_8$ as a product $\tilde{\mathbf{E}}_8 \mathbf{P}_8^{(\pi_8)}$, the Equation (8) will take the form:

$$\mathbf{Y}_{8\times 1} = \mathbf{W}_8^{(0)}(\mathbf{E}_4 \oplus \mathbf{Q}_4)\mathbf{P}_8^{(\pi_8)}\mathbf{X}_{8\times 1} \tag{10}$$

where

$$\mathbf{W}_8^{(0)} = \mathbf{H}_2 \otimes \mathbf{I}_4.$$

Permute columns of the matrix $\mathbf{E}_4$ and rows of the matrix $\mathbf{Q}_4$ according to permutation $\pi_4$. As a result of such permutations, we obtain the matrices

$$\tilde{\mathbf{E}}_4 = \left[ \begin{array}{c:c} \mathbf{A}_2 & \mathbf{B}_2 \\ \hdashline \mathbf{A}_2 & -\mathbf{B}_2 \end{array} \right] \quad \text{and} \quad \tilde{\mathbf{Q}}_4 = \left[ \begin{array}{c:c} \mathbf{C}_2 & \mathbf{C}_2 \\ \hdashline \mathbf{D}_2 & -\mathbf{D}_2 \end{array} \right]$$

where

$$\mathbf{A}_2 = \left[ \begin{array}{c:c} a_8 & a_8 \\ \hdashline a_8 & -a_8 \end{array} \right], \quad \mathbf{B}_2 = \left[ \begin{array}{c:c} a_8 & a_8 \\ \hdashline c_8 & -c_8 \end{array} \right],$$

$$\mathbf{C}_2 = \left[ \begin{array}{c:c} a_8 & a_8 \\ \hdashline c_8 & -c_8 \end{array} \right], \quad \mathbf{D}_2 = \left[ \begin{array}{c:c} b_8 & d_8 \\ \hdashline d_8 & b_8 \end{array} \right].$$

Such structures of the matrices $\tilde{\mathbf{E}}_4$ and $\tilde{\mathbf{Q}}_4$ allow to apply the same schemes of factorization. Therefore, we can write

$$\mathbf{Y}_{8\times 1} = \mathbf{W}_8^{(0)}\tilde{\mathbf{W}}_8^{(0)}(\mathbf{A}_2 \oplus \mathbf{B}_2 \oplus \mathbf{C}_2 \oplus \mathbf{D}_2)\tilde{\mathbf{W}}_8^{(1)}\mathbf{P}_8^{(\pi_8)}\mathbf{X}_{8\times 1} \tag{11}$$

where

$$\tilde{\mathbf{W}}_8^{(0)} = (\mathbf{H}_2 \otimes \mathbf{I}_2) \oplus \mathbf{P}_4^{(\pi_4)}, \qquad \tilde{\mathbf{W}}_8^{(1)} = \mathbf{P}_4^{(\pi_4)} \oplus (\mathbf{H}_2 \otimes \mathbf{I}_2),$$

For matrices $\mathbf{A}_2, \mathbf{B}_2, \mathbf{C}_2, \mathbf{D}_2$ we can offer the following factorization schemes leading to a reduction in computational complexity:

$$\mathbf{A}_2 = \begin{bmatrix} a_8 & a_8 \\ a_8 & -a_8 \end{bmatrix} = (a_8 \oplus a_8)\mathbf{H}_2, \quad \mathbf{B}_2 = \begin{bmatrix} a_8 & a_8 \\ c_8 & -c_8 \end{bmatrix} = (a_8 \oplus c_8)\mathbf{H}_2,$$

$$\mathbf{C}_2 = \begin{bmatrix} a_8 & a_8 \\ c_8 & -c_8 \end{bmatrix} = (a_8 \oplus c_8)\mathbf{H}_2, \qquad \mathbf{D}_2 = \begin{bmatrix} b_8 & d_8 \\ d_8 & b_8 \end{bmatrix} = \mathbf{H}_2\frac{1}{2}[(b_8 + d_8) \oplus (b_8 - d_8)]\mathbf{H}_2$$

Taking into account the above factorization schemes, we can finally write

$$\mathbf{Y}_{8\times 1} = \mathbf{W}_8^{(0)}\tilde{\mathbf{W}}_8^{(0)}\mathbf{W}_8^{(3)}\mathbf{D}_8\mathbf{W}_8^{(4)}\tilde{\mathbf{W}}_8^{(1)}\mathbf{P}_8^{(\pi_8)}\mathbf{X}_{8\times 1} \tag{12}$$

where

$$\mathbf{W}_8^{(4)} = \mathbf{I}_4 \otimes \mathbf{H}_2, \qquad \mathbf{W}_8^{(3)} = \mathbf{I}_6 \oplus \mathbf{H}_2,$$

$$\mathbf{D}_8 = diag(\varphi_0, \varphi_1, \varphi_2, \varphi_3, \varphi_4, \varphi_5, \varphi_6, \varphi_7),$$

$$\varphi_0, \varphi_1, \varphi_2, \varphi_4 = a_8 = 1, \varphi_3, \varphi_5 = e_8 = -j, \varphi_6 = -j0.7071, \varphi_7 = 0.7071.$$

Expression (12) describes the Winograd-type fast Fourier transform algorithm for $N = 8$. Figure 2 shows a data flow graph of synthesized algorithm for 8 point DFT. As can be seen, in this case the algorithm takes 2 multiplications and 26 additions.

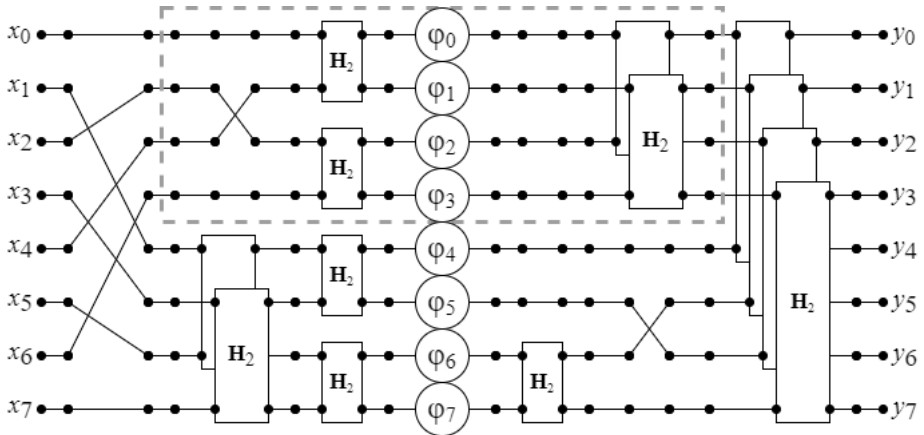

**Figure 2.** The data flow graph of the proposed algorithm for computation of 8-point DFT.

*4.3. Fast DFT Algorithm for N = 16*

Now let us consider the synthesis of a similar algorithm for $N = 16$. In matrix–vector notation, we can rewrite the DFT in the following form:

$$\mathbf{Y}_{16\times 1} = \mathbf{E}_{16}\mathbf{X}_{16\times 1} \tag{13}$$

where

$$\mathbf{X}_{16\times 1} = [x_0, x_1, x_2, x_3, x_4, x_5, x_6, x_7, x_8, x_9, x_{10}, x_{11}, x_{12}, x_{13}, x_{14}, x_{15}]^{\mathrm{T}}$$

$$\mathbf{Y}_{16\times 1} = [y_0, y_1, y_2, y_3, y_4, y_5, y_6, y_7, y_8, y_9, y_{10}, y_{11}, y_{12}, y_{13}, y_{14}, y_{15}]^{\mathrm{T}}$$

$$\mathbf{E}_{16} = \left[ \begin{array}{c:c} \mathbf{E}_8^{(0,0)} & \mathbf{E}_8^{(0,1)} \\ \hdashline \mathbf{E}_8^{(1,0)} & -\mathbf{E}_8^{(1,1)} \end{array} \right].$$

where

$$\mathbf{E}_8^{(0,0)} = \left[ \begin{array}{cccc:cccc}
a_{16} & a_{16} & a_{16} & a_{16} & a_{16} & a_{16} & a_{16} & a_{16} \\
a_{16} & b_{16} & c_{16} & d_{16} & e_{16} & f_{16} & g_{16} & h_{16} \\
a_{16} & c_{16} & e_{16} & g_{16} & -a_{16} & -c_{16} & -e_{16} & -g_{16} \\
a_{16} & d_{16} & g_{16} & -b_{16} & -e_{16} & -h_{16} & c_{16} & f_{16} \\ \hdashline
a_{16} & e_{16} & -a_{16} & -e_{16} & a_{16} & e_{16} & -a_{16} & -e_{16} \\
a_{16} & f_{16} & -c_{16} & -h_{16} & e_{16} & -b_{16} & -g_{16} & d_{16} \\
a_{16} & g_{16} & -e_{16} & c_{16} & -a_{16} & -g_{16} & e_{16} & -c_{16} \\
a_{16} & h_{16} & -g_{16} & f_{16} & -e_{16} & d_{16} & -c_{16} & b_{16}
\end{array} \right],$$

$$\mathbf{E}_8^{(0,1)} = \left[ \begin{array}{cccc:cccc}
a_{16} & a_{16} & a_{16} & a_{16} & a_{16} & a_{16} & a_{16} & a_{16} \\
-a_{16} & -b_{16} & -c_{16} & -d_{16} & -e_{16} & -f_{16} & -g_{16} & -h_{16} \\
a_{16} & c_{16} & e_{16} & g_{16} & -a_{16} & -c_{16} & -e_{16} & -g_{16} \\
-a_{16} & -d_{16} & -g_{16} & b_{16} & e_{16} & h_{16} & -c_{16} & -f_{16} \\ \hdashline
a_{16} & e_{16} & -a_{16} & -e_{16} & a_{16} & e_{16} & -a_{16} & -e_{16} \\
-a_{16} & -f_{16} & c_{16} & h_{16} & -e_{16} & b_{16} & g_{16} & -d_{16} \\
a_{16} & g_{16} & -e_{16} & c_{16} & -a_{16} & -g_{16} & e_{16} & -c_{16} \\
-a_{16} & -h_{16} & g_{16} & -f_{16} & e_{16} & -d_{16} & c_{16} & -b_{16}
\end{array} \right],$$

$$\mathbf{E}_8^{(1,0)} = \left[ \begin{array}{cccc:cccc}
a_{16} & -a_{16} & a_{16} & -a_{16} & a_{16} & -a_{16} & a_{16} & -a_{16} \\
a_{16} & -b_{16} & c_{16} & -d_{16} & e_{16} & -f_{16} & g_{16} & -h_{16} \\
a_{16} & -c_{16} & e_{16} & -g_{16} & -a_{16} & c_{16} & -e_{16} & g_{16} \\
a_{16} & -d_{16} & g_{16} & b_{16} & -e_{16} & h_{16} & c_{16} & -f_{16} \\ \hdashline
a_{16} & -e_{16} & -a_{16} & e_{16} & a_{16} & -e_{16} & -a_{16} & e_{16} \\
a_{16} & -f_{16} & -c_{16} & h_{16} & e_{16} & b_{16} & -g_{16} & -d_{16} \\
a_{16} & -g_{16} & -e_{16} & -c_{16} & -a_{16} & g_{16} & e_{16} & c_{16} \\
a_{16} & -h_{16} & -g_{16} & -f_{16} & -e_{16} & -d_{16} & -c_{16} & -b_{16}
\end{array} \right],$$

$$\mathbf{E}_8^{(1,1)} = \left[ \begin{array}{cccc:cccc}
a_{16} & -a_{16} & a_{16} & -a_{16} & a_{16} & -a_{16} & a_{16} & -a_{16} \\
-a_{16} & b_{16} & -c_{16} & d_{16} & -e_{16} & f_{16} & -g_{16} & h_{16} \\
a_{16} & -c_{16} & e_{16} & -g_{16} & -a_{16} & c_{16} & -e_{16} & g_{16} \\
-a_{16} & d_{16} & -g_{16} & -b_{16} & e_{16} & -h_{16} & -c_{16} & f_{16} \\ \hdashline
a_{16} & -e_{16} & -a_{16} & e_{16} & a_{16} & -e_{16} & -a_{16} & e_{16} \\
-a_{16} & f_{16} & c_{16} & -h_{16} & -e_{16} & -b_{16} & g_{16} & d_{16} \\
a_{16} & -g_{16} & -e_{16} & -c_{16} & -a_{16} & g_{16} & e_{16} & c_{16} \\
-a_{16} & h_{16} & g_{16} & f_{16} & e_{16} & d_{16} & c_{16} & b_{16}
\end{array} \right].$$

where

$$a_{16} = 1, \qquad b_{16} = 0.9239 - j0.3827, \qquad c_{16} = 0.7071 - j0.7071,$$

$$d_{16} = 0.3827 - j0.9239, \qquad e_{16} = -j, \qquad f_{16} = -0.3827 - j0.9239,$$

$$g_{16} = -0.7071 - j0.7071, \qquad h_{16} = -0.9239 - j0.3827.$$

Let us define the permutation $\pi_{16}$ in the following form:

$$\pi_{16} = \left( \begin{array}{cccccccc:cccccccc}
1 & 2 & 3 & 4 & 5 & 6 & 7 & 8 & 9 & 10 & 11 & 12 & 13 & 14 & 15 & 16 \\
1 & 3 & 5 & 7 & 9 & 11 & 13 & 15 & 2 & 4 & 6 & 8 & 10 & 12 & 14 & 16
\end{array} \right)$$

Permute columns of the matrix $\mathbf{E}_{16}$ according to permutation $\pi_{16}$. As a result of such permutation, we obtain the matrix

$$\tilde{\mathbf{E}}_{16} = \left[ \begin{array}{c|c} \mathbf{E}_8 & \mathbf{Q}_8 \\ \hline \mathbf{E}_8 & -\mathbf{Q}_8 \end{array} \right] = \tilde{\mathbf{E}}_{16}\mathbf{P}_{16}^{(\pi_{16})}$$

where

$$\mathbf{E}_8 = \left[ \begin{array}{cccc|cccc}
a_{16} & a_{16} & a_{16} & a_{16} & a_{16} & a_{16} & a_{16} & a_{16} \\
a_{16} & c_{16} & e_{16} & g_{16} & -a_{16} & -c_{16} & -e_{16} & -g_{16} \\
a_{16} & e_{16} & -a_{16} & -e_{16} & a_{16} & a_{16} & -a_{16} & -e_{16} \\
a_{16} & g_{16} & -e_{16} & c_{16} & -a_{16} & -g_{16} & e_{16} & -c_{16} \\
\hline
a_{16} & -a_{16} & a_{16} & -a_{16} & a_{16} & -a_{16} & a_{16} & -a_{16} \\
a_{16} & -c_{16} & e_{16} & -g_{16} & -a_{16} & c_{16} & -e_{16} & g_{16} \\
a_{16} & -e_{16} & -a_{16} & e_{16} & a_{16} & -e_{16} & -a_{16} & e_{16} \\
a_{16} & -g_{16} & -e_{16} & -c_{16} & -a_{16} & g_{16} & e_{16} & c_{16}
\end{array} \right],$$

$$\mathbf{Q}_8 = \left[ \begin{array}{cccc|cccc}
a_{16} & a_{16} & a_{16} & a_{16} & a_{16} & a_{16} & a_{16} & a_{16} \\
b_{16} & d_{16} & f_{16} & h_{16} & -b_{16} & -d_{16} & -f_{16} & -h_{16} \\
c_{16} & g_{16} & -c_{16} & -g_{16} & c_{16} & g_{16} & -c_{16} & -g_{16} \\
d_{16} & -b_{16} & -h_{16} & f_{16} & -d_{16} & b_{16} & h_{16} & -f_{16} \\
\hline
e_{16} & -e_{16} & e_{16} & -e_{16} & e_{16} & -e_{16} & e_{16} & -e_{16} \\
f_{16} & -h_{16} & -b_{16} & d_{16} & -f_{16} & h_{16} & b_{16} & -d_{16} \\
g_{16} & c_{16} & -g_{16} & -c_{16} & g_{16} & c_{16} & -g_{16} & -c_{16} \\
h_{16} & f_{16} & d_{16} & b_{16} & -h_{16} & -f_{16} & -d_{16} & -b_{16}
\end{array} \right].$$

According to the concept, expression (4) for a given transform size can be rewritten as follows:

$$\mathbf{Y}_{16\times1} = (\mathbf{H}_2 \otimes \mathbf{I}_8)(\mathbf{E}_8 \oplus \mathbf{Q}_8)\mathbf{P}_{16}^{(\pi_{16})}\mathbf{X}_{16\times1} \qquad (14)$$

Such a matrix structure allows for the factorization of the matrix $\tilde{\mathbf{E}}_{16}$; similarly, as was done in the case of the matrix of order $N = 8$ [19].

If we write the matrix $\mathbf{E}_{16}$ as a product $\tilde{\mathbf{E}}_{16}\mathbf{P}_{16}^{(\pi_{16})}$, Equation (13) takes the form

$$\mathbf{Y}_{16\times1} = \mathbf{W}_{16}^{(0)}(\mathbf{E}_8 \oplus \mathbf{Q}_8)\mathbf{P}_{16}^{(\pi_{16})}\mathbf{X}_{16\times1} \qquad (15)$$

where the corresponding permutation matrix $\mathbf{P}_{16}^{(\pi_{16})}$ takes the following form:

$$\check{\mathbf{P}}_{4\times8} = \left[ \begin{array}{cccc|cccc} 1 & & & & & & & \\ & & 1 & & & & & \\ \hline & & & & 1 & & & \\ & & & & & & 1 & \end{array} \right], \qquad \hat{\mathbf{P}}_{4\times8} = \left[ \begin{array}{cccc|cccc} & 1 & & & & & & \\ & & & 1 & & & & \\ \hline & & & & & 1 & & \\ & & & & & & & 1 \end{array} \right],$$

$$\check{\mathbf{P}}_{8\times16} = \check{\mathbf{P}}_{4\times8} \oplus \check{\mathbf{P}}_{4\times8}, \quad \hat{\mathbf{P}}_{8\times16} = \hat{\mathbf{P}}_{4\times8} \oplus \hat{\mathbf{P}}_{4\times8},$$

$$\mathbf{P}_{16}^{(\pi_{16})} = \mathbf{P}_{16}^{(0)} = \left[ \begin{array}{c} \check{\mathbf{P}}_{8\times16} \\ \hline \hat{\mathbf{P}}_{8\times16} \end{array} \right] \quad \text{and} \quad \mathbf{W}_{16}^{(0)} = \mathbf{H}_2 \otimes \mathbf{I}_8.$$

Now let us permute columns of the matrix $\mathbf{E}_8$ according to permutation $\pi_8$. As a result of such a permutation, we obtain the matrix

$$\tilde{\mathbf{E}}_8 = \left[\begin{array}{cccc:cccc} a_{16} & a_{16} & a_{16} & a_{16} & a_{16} & a_{16} & a_{16} & a_{16} \\ a_{16} & e_{16} & -a_{16} & -e_{16} & e_{16} & g_{16} & -c_{16} & -g_{16} \\ a_{16} & -a_{16} & a_{16} & -a_{16} & e_{16} & -e_{16} & e_{16} & -e_{16} \\ a_{16} & -e_{16} & -a_{16} & e_{16} & g_{16} & c_{16} & -g_{16} & -c_{16} \\ \hdashline a_{16} & a_{16} & a_{16} & a_{16} & -a_{16} & -a_{16} & -a_{16} & -a_{16} \\ a_{16} & e_{16} & -a_{16} & -e_{16} & -c_{16} & -g_{16} & e_{16} & g_{16} \\ a_{16} & -a_{16} & a_{16} & -a_{16} & -e_{16} & e_{16} & -e_{16} & e_{16} \\ a_{16} & -e_{16} & -a_{16} & e_{16} & -g_{16} & -c_{16} & g_{16} & c_{16} \end{array}\right] = \left[\begin{array}{c:c} \mathbf{A}_4 & \mathbf{B}_4 \\ \hdashline \mathbf{A}_4 & -\mathbf{B}_4 \end{array}\right],$$

where

$$\mathbf{A}_4 = \left[\begin{array}{cccc} a_{16} & a_{16} & a_{16} & a_{16} \\ a_{16} & e_{16} & -a_{16} & -e_{16} \\ a_{16} & -a_{16} & a_{16} & -a_{16} \\ a_{16} & -e_{16} & -a_{16} & e_{16} \end{array}\right] \quad \text{and} \quad \mathbf{B}_4 = \left[\begin{array}{cccc} a_{16} & a_{16} & a_{16} & a_{16} \\ c_{16} & g_{16} & -c_{16} & -g_{16} \\ e_{16} & -e_{16} & e_{16} & -e_{16} \\ g_{16} & c_{16} & -g_{16} & -c_{16} \end{array}\right].$$

Then the matrix $\mathbf{E}_8$ can be represented as a product $\tilde{\mathbf{E}}_8 \mathbf{P}_8^{(\pi_8)} = \mathbf{W}_8^{(0)}(\mathbf{A}_4 \oplus \mathbf{B}_4)\mathbf{P}_8^{(\pi_8)}$. Next, we permute rows of the matrix $\mathbf{Q}_8$ according to permutation $\pi_8$. As a result of such a permutation, we obtain the matrix $\tilde{\mathbf{Q}}_8$

$$\tilde{\mathbf{Q}}_8 = \left[\begin{array}{cccc:cccc} a_{16} & a_{16} & a_{16} & a_{16} & a_{16} & a_{16} & a_{16} & a_{16} \\ c_{16} & g_{16} & -c_{16} & -g_{16} & c_{16} & g_{16} & -c_{16} & -g_{16} \\ e_{16} & -e_{16} & e_{16} & -e_{16} & e_{16} & -e_{16} & e_{16} & -e_{16} \\ g_{16} & c_{16} & -g_{16} & -c_{16} & g_{16} & c_{16} & -g_{16} & -c_{16} \\ \hdashline b_{16} & d_{16} & f_{16} & h_{16} & -b_{16} & -d_{16} & -f_{16} & -h_{16} \\ d_{16} & -b_{16} & -h_{16} & f_{16} & -d_{16} & b_{16} & h_{16} & -f_{16} \\ f_{16} & -h_{16} & -b_{16} & d_{16} & -f_{16} & h_{16} & b_{16} & -d_{16} \\ h_{16} & f_{16} & d_{16} & b_{16} & -h_{16} & -f_{16} & -d_{16} & -b_{16} \end{array}\right] = \left[\begin{array}{c:c} \mathbf{C}_4 & \mathbf{C}_4 \\ \hdashline \mathbf{D}_4 & -\mathbf{D}_4 \end{array}\right],$$

where

$$\mathbf{C}_4 = \left[\begin{array}{cccc} a_{16} & a_{16} & a_{16} & a_{16} \\ c_{16} & g_{16} & -c_{16} & -g_{16} \\ e_{16} & -e_{16} & e_{16} & -e_{16} \\ g_{16} & c_{16} & -g_{16} & -c_{16} \end{array}\right] \quad \text{and} \quad \mathbf{D}_4 = \left[\begin{array}{cccc} b_{16} & d_{16} & f_{16} & h_{16} \\ d_{16} & -b_{16} & -h_{16} & f_{16} \\ f_{16} & -h_{16} & -b_{16} & d_{16} \\ h_{16} & f_{16} & d_{16} & b_{16} \end{array}\right].$$

Then the matrix $\mathbf{Q}_8$ can be represented as a product $\mathbf{P}_8^{(\pi_8)}\tilde{\mathbf{Q}}_8 = \mathbf{P}_8^{(\pi_8)}(\mathbf{C}_4 \oplus \mathbf{D}_4)\mathbf{W}_8^{(0)}$. Taking into account the above factorization schemes, we can finally write

$$\mathbf{Y}_{16\times 1} = \mathbf{W}_{16}^{(0)}\tilde{\mathbf{W}}_{16}^{(0)}(\mathbf{A}_4 \oplus \mathbf{B}_4 \oplus \mathbf{C}_4 \oplus \mathbf{D}_4)\tilde{\mathbf{W}}_{16}^{(1)}\mathbf{P}_{16}^{(0)}\mathbf{X}_{16\times 1} \tag{16}$$

where

$$\tilde{\mathbf{W}}_{16}^{(0)} = \mathbf{W}_8^{(0)} \oplus \left[\mathbf{P}_8^{(\pi_8)}\right]^{\mathrm{T}}, \qquad \tilde{\mathbf{W}}_{16}^{(1)} = \mathbf{P}_8^{(\pi_8)} \oplus \mathbf{W}_8^{(0)}.$$

We now consider the matrices $\mathbf{A}_4, \mathbf{B}_4, \mathbf{C}_4$, and $\mathbf{D}_4$. Permute columns of the matrix $\mathbf{A}_4$ according to permutation $\pi_4$. As a result of such a permutation, we obtain the matrix

$$\tilde{\mathbf{A}}_4 = \left[\begin{array}{cc:cc} a_{16} & a_{16} & a_{16} & a_{16} \\ a_{16} & -a_{16} & e_{16} & -e_{16} \\ \hdashline a_{16} & a_{16} & -a_{16} & -a_{16} \\ a_{16} & -a_{16} & -e_{16} & e_{16} \end{array}\right] = \left[\begin{array}{c:c} \mathbf{A}_2 & \mathbf{B}_2 \\ \hdashline \mathbf{A}_2 & -\mathbf{B}_2 \end{array}\right].$$

Next, we permute rows of the matrix $\mathbf{B}_4$ according to permutation $\pi_4$. As a result of such a permutation, we obtain the matrix $\tilde{\mathbf{B}}_4$

$$\tilde{\mathbf{B}}_4 = \left[ \begin{array}{cc:cc} a_{16} & a_{16} & a_{16} & a_{16} \\ e_{16} & -e_{16} & e_{16} & -e_{16} \\ \hdashline c_{16} & g_{16} & -c_{16} & -g_{16} \\ g_{16} & c_{16} & -g_{16} & -c_{16} \end{array} \right] = \left[ \begin{array}{c:c} \mathbf{C}_2 & \mathbf{C}_2 \\ \hdashline \mathbf{D}_2 & -\mathbf{D}_2 \end{array} \right].$$

Now, we permute rows of the matrix $\mathbf{C}_4$ according to permutation $\pi_4$. As a result of such a permutation, we obtain the matrix $\tilde{\mathbf{C}}_4$

$$\tilde{\mathbf{C}}_4 = \left[ \begin{array}{cc:cc} a_{16} & a_{16} & a_{16} & a_{16} \\ e_{16} & -e_{16} & e_{16} & -e_{16} \\ \hdashline c_{16} & g_{16} & -c_{16} & -g_{16} \\ g_{16} & c_{16} & -g_{16} & -c_{16} \end{array} \right] = \left[ \begin{array}{c:c} \mathbf{F}_2 & \mathbf{F}_2 \\ \hdashline \mathbf{G}_2 & -\mathbf{G}_2 \end{array} \right].$$

Next, we define the permutation in the following way:

$$\tilde{\pi}_4 = \left( \begin{array}{cccc} 1 & 2 & 3 & 4 \\ 1 & 2 & 4 & 3 \end{array} \right)$$

Permute rows and columns of the matrix $\mathbf{D}_4$ according to permutation $\tilde{\pi}_4$. As a result of such a permutation, we obtain the matrix $\tilde{\mathbf{D}}_4$.

$$\tilde{\mathbf{D}}_4 = \left[ \begin{array}{cc:cc} b_{16} & d_{16} & h_{16} & f_{16} \\ d_{16} & -b_{16} & f_{16} & -h_{16} \\ \hdashline h_{16} & f_{16} & b_{16} & d_{16} \\ f_{16} & -h_{16} & d_{16} & -b_{16} \end{array} \right] = \mathbf{P}_4^{(\tilde{\pi}_4)} \mathbf{D}_4 \mathbf{P}_4^{(\tilde{\pi}_4)} = \left[ \begin{array}{c:c} \tilde{\mathbf{J}}_2 & \tilde{\mathbf{K}}_2 \\ \hdashline \tilde{\mathbf{K}}_2 & -\tilde{\mathbf{J}}_2 \end{array} \right]$$

where

$$\mathbf{P}_4^{(\tilde{\pi}_4)} = \left[ \begin{array}{cc:cc} 1 & & & \\ & 1 & & \\ \hdashline & & & 1 \\ & & 1 & \end{array} \right].$$

Then

$$\mathbf{D}_4 = \mathbf{P}_4^{(\tilde{\pi}_4)} (\mathbf{H}_2 \otimes \mathbf{I}_2) \frac{1}{2} \left[ (\tilde{\mathbf{J}}_2 + \tilde{\mathbf{K}}_2) \oplus (\tilde{\mathbf{J}}_2 - \tilde{\mathbf{K}}_2) \right] (\mathbf{H}_2 \otimes \mathbf{I}_2) \mathbf{P}_4^{(\tilde{\pi}_4)}$$

where

$$\frac{1}{2} (\tilde{\mathbf{J}}_2 + \tilde{\mathbf{K}}_2) = \frac{1}{2} \left[ \begin{array}{c:c} b_{16} + h_{16} & d_{16} + f_{16} \\ \hdashline d_{16} + f_{16} & -(b_{16} + h_{16}) \end{array} \right] = \mathbf{J}_2,$$

$$\frac{1}{2} (\tilde{\mathbf{J}}_2 - \tilde{\mathbf{K}}_2) = \frac{1}{2} \left[ \begin{array}{c:c} b_{16} - h_{16} & d_{16} - f_{16} \\ \hdashline d_{16} - f_{16} & -(b_{16} - h_{16}) \end{array} \right] = \mathbf{K}_2.$$

Taking into account the matrix transformations performed above, we can write

$$\mathbf{Y}_{16 \times 1} = \mathbf{W}_{16}^{(0)} \tilde{\mathbf{W}}_{16}^{(0)} \mathbf{W}_{16}^{(4)} \mathbf{P}_{16}^{(4)} \mathbf{D}_{16} \mathbf{P}_{16}^{(3)} \mathbf{W}_{16}^{(3)} \tilde{\mathbf{W}}_{16}^{(1)} \mathbf{P}_{16}^{(0)} \mathbf{X}_{16 \times 1} \tag{17}$$

where

$$\mathbf{D}_{16} = \mathbf{A}_2 \oplus \mathbf{B}_2 \oplus \mathbf{C}_2 \oplus \mathbf{D}_2 \oplus \mathbf{F}_2 \oplus \mathbf{G}_2 \oplus \mathbf{J}_2 \oplus \mathbf{K}_2,$$

$$\mathbf{P}_{16}^{(4)} = \mathbf{I}_4 \oplus \mathbf{P}_4^{(\pi_4)} \oplus \mathbf{P}_4^{(\pi_4)} \oplus \mathbf{P}_4^{(\tilde{\pi}_4)}, \qquad \mathbf{W}_{16}^{(4)} = \mathbf{W}_4^{(0)} \oplus \mathbf{I}_8 \oplus \mathbf{W}_4^{(0)},$$

$$\mathbf{W}_{16}^{(3)} = \mathbf{I}_4 \oplus \mathbf{W}_4^{(0)} \oplus \mathbf{W}_4^{(0)} \oplus \mathbf{W}_4^{(0)}, \qquad \mathbf{P}_{16}^{(3)} = \mathbf{P}_4^{(\pi_4)} \oplus \mathbf{I}_8 \oplus \mathbf{P}_4^{(\tilde{\pi}_4)},$$

$$\mathbf{A}_2 = \left[\begin{array}{c:c} a_{16} & a_{16} \\ \hline a_{16} & -a_{16} \end{array}\right], \quad \mathbf{B}_2 = \left[\begin{array}{c:c} a_{16} & a_{16} \\ \hline e_{16} & -e_{16} \end{array}\right],$$

$$\mathbf{C}_2 = \mathbf{F}_2 = \left[\begin{array}{c:c} a_{16} & a_{16} \\ \hline e_{16} & -e_{16} \end{array}\right], \quad \mathbf{D}_2 = \mathbf{G}_2 = \left[\begin{array}{c:c} c_{16} & g_{16} \\ \hline g_{16} & c_{16} \end{array}\right],$$

$$\mathbf{J}_2 = \frac{1}{2}\left[\begin{array}{c:c} b_{16}+h_{16} & d_{16}+f_{16} \\ \hline d_{16}+f_{16} & -(b_{16}+h_{16}) \end{array}\right], \quad \mathbf{K}_2 = \frac{1}{2}\left[\begin{array}{c:c} b_{16}-h_{16} & d_{16}-f_{16} \\ \hline d_{16}-f_{16} & -(b_{16}-h_{16}) \end{array}\right].$$

In turn, the matrices $\mathbf{A}_2$, $\mathbf{B}_2$, $\mathbf{C}_2$, $\mathbf{D}_2$, $\mathbf{F}_2$, $\mathbf{G}_2$, $\mathbf{J}_2$ and $\mathbf{K}_2$ also have structures that provide effective factorization, which leads to a decrease in the multiplicative complexity of calculations:

$$\mathbf{A}_2 = \left[\begin{array}{c:c} a_{16} & a_{16} \\ \hline a_{16} & -a_{16} \end{array}\right] = (a_{16} \oplus a_{16})\mathbf{H}_2,$$

$$\mathbf{B}_2 = \left[\begin{array}{c:c} a_{16} & a_{16} \\ \hline e_{16} & -e_{16} \end{array}\right] = (a_{16} \oplus e_{16})\mathbf{H}_2,$$

$$\mathbf{C}_2 = \mathbf{F}_2 \left[\begin{array}{c:c} a_{16} & a_{16} \\ \hline e_{16} & -e_{16} \end{array}\right] = (a_{16} \oplus e_{16})\mathbf{H}_2,$$

$$\mathbf{D}_2 = \mathbf{G}_2 \left[\begin{array}{c:c} c_{16} & g_{16} \\ \hline g_{16} & c_{16} \end{array}\right] = \mathbf{H}_2 \frac{1}{2}[(c_{16}+g_{16}) \oplus (c_{16}-g_{16})]\mathbf{H}_2,$$

$$\mathbf{J}_2 = \frac{1}{2}\left[\begin{array}{c:c} b_{16}+h_{16} & d_{16}+f_{16} \\ \hline d_{16}+f_{16} & -(b_{16}+h_{16}) \end{array}\right] = \mathbf{T}_{2\times3}\frac{1}{2}\left[\begin{array}{c:c:c} j_{21} & 0 & 0 \\ \hline 0 & j_{22} & 0 \\ \hline 0 & 0 & j_{23} \end{array}\right]\mathbf{T}_{3\times2},$$

where

$$\mathbf{T}_{2\times3} = \left[\begin{array}{ccc} 1 & 0 & 1 \\ 0 & 1 & 1 \end{array}\right], \quad \mathbf{T}_{3\times2} = \left[\begin{array}{cc} 1 & 0 \\ 0 & 1 \\ 1 & 1 \end{array}\right],$$

$$j_{21} = (b_{16}+h_{16}) - (d_{16}+f_{16}),$$

$$j_{22} = -[(b_{16}+h_{16}) + (d_{16}+f_{16})],$$

$$j_{23} = d_{16}+f_{16}.$$

$$\mathbf{K}_2 = \frac{1}{2}\left[\begin{array}{c:c} b_{16}-h_{16} & d_{16}-f_{16} \\ \hline d_{16}-f_{16} & -(b_{16}-h_{16}) \end{array}\right] = \mathbf{T}_{2\times3}\frac{1}{2}\left[\begin{array}{c:c:c} k_{21} & 0 & 0 \\ \hline 0 & k_{22} & 0 \\ \hline 0 & 0 & k_{23} \end{array}\right]\mathbf{T}_{3\times2}.$$

where

$$k_{21} = (b_{16}-h_{16}) - (d_{16}-f_{16}),$$

$$k_{22} = -[(b_{16}-h_{16}) + (d_{16}-f_{16})],$$

$$k_{23} = (d_{16}-f_{16}).$$

Combining the above partial decompositions in a single procedure, we can rewrite (13) as follows:

$$\mathbf{Y}_{16\times1} = \mathbf{W}_{16}^{(0)}\tilde{\mathbf{W}}_{16}^{(0)}\mathbf{W}_{16}^{(4)}\mathbf{P}_{16}^{(4)}\mathbf{W}_{16}^{(6)}\mathbf{A}_{16\times18}\mathbf{D}_{18}\mathbf{A}_{18\times16}\mathbf{W}_{16}^{(5)}\mathbf{P}_{16}^{(3)}\mathbf{W}_{16}^{(3)}\tilde{\mathbf{W}}_{16}^{(1)}\mathbf{P}_{16}^{(0)}\mathbf{X}_{16\times1} \qquad (18)$$

where

$$\mathbf{W}_{16}^{(5)} = (\mathbf{I}_6 \otimes \mathbf{H}_2) \oplus \mathbf{I}_4, \qquad \mathbf{W}_{16}^{(6)} = \mathbf{I}_6 \oplus \mathbf{H}_2 \oplus \mathbf{I}_2 \oplus \mathbf{H}_2 \oplus \mathbf{I}_4,$$

$$\mathbf{D}_{18} = diag(\varphi_0, \varphi_1, \ldots, \varphi_{17}),$$

$$\varphi_0, \varphi_1, \varphi_2, \varphi_4, \varphi_8 = a_{16} = 1, \qquad \varphi_3, \varphi_5, \varphi_9 = e_{16} = -j,$$

$$\varphi_6, \varphi_{10} = \frac{1}{2}(c_{16} + g_{16}) = -j0.7071, \qquad \varphi_7, \varphi_{11} = \frac{1}{2}(c_{16} - g_{16}) = 0.7071,$$

$$\varphi_{12} = \frac{1}{2}[(b_{16} + h_{16}) - (d_{16} + f_{16})] = j0.5412,$$

$$\varphi_{13} = -\frac{1}{2}[(b_{16} + h_{16}) - (d_{16} + f_{16})] = j1.3066,$$

$$\varphi_{14} = \frac{1}{2}(d_{16} + f_{16}) = -j0.9239,$$

$$\varphi_{15} = \frac{1}{2}[(b_{16} - h_{16}) - (d_{16} - f_{16})] = -0.5412,$$

$$\varphi_{16} = -\frac{1}{2}[(b_{16} - h_{16}) + (d_{16} - f_{16})] = 1.3066,$$

$$\varphi_{17} = \frac{1}{2}(d_{16} - f_{16}) = -0.3827,$$

$$\mathbf{A}_{16 \times 18} = \mathbf{I}_{12} \oplus \mathbf{T}_{2 \times 3} \oplus \mathbf{T}_{2 \times 3}, \qquad \mathbf{A}_{18 \times 16} = \mathbf{I}_{12} \oplus \mathbf{T}_{3 \times 2} \oplus \mathbf{T}_{3 \times 2}.$$

Figure 3 shows a data flow graph of synthesized algorithm for 16 point DFT. As can be seen, in this case the algorithm takes 10 multiplications and 74 additions.

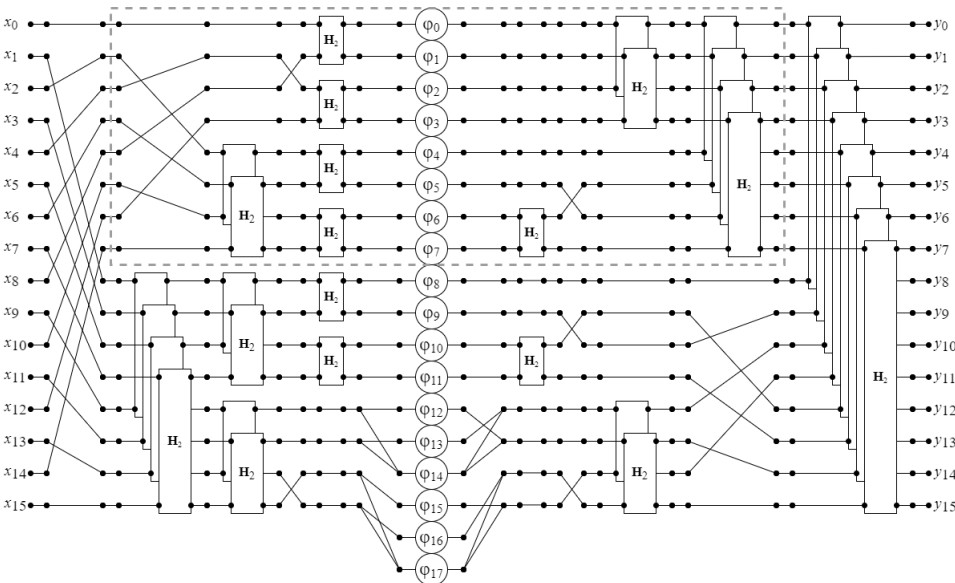

**Figure 3.** The data flow graph of the proposed algorithm for computation of 16-point DFT.

*4.4. Fast DFT Algorithm for N = 32*

Now let us consider the synthesis of a similar algorithm for *N* = 32. In matrix–vector notation, we can rewrite the DFT in the following form:

$$\mathbf{Y}_{32 \times 1} = \mathbf{E}_{32} \mathbf{X}_{32 \times 1} \tag{19}$$

where

$$\mathbf{X}_{32 \times 1} = [x_0, x_1, x_2, \ldots, x_{29}, x_{30}, x_{31}]^{\mathrm{T}}, \quad \mathbf{Y}_{32 \times 1} = [y_0, y_1, y_2, \ldots, y_{29}, y_{30}, y_{31}]^{\mathrm{T}}$$

Let us define the permutation $\pi_{32}$ in the following form:

$$\pi_{32} = \left( \begin{array}{cccccccc|cccccccc} 1 & 2 & 3 & 4 & 5 & 6 & 7 & 8 & 9 & 10 & 11 & 12 & 13 & 14 & 15 & 16 \\ 1 & 3 & 5 & 7 & 9 & 11 & 13 & 15 & 17 & 19 & 21 & 23 & 25 & 27 & 29 & 31 \\ \\ 17 & 18 & 19 & 20 & 21 & 22 & 23 & 24 & 25 & 26 & 27 & 28 & 29 & 30 & 31 & 32 \\ 2 & 4 & 6 & 8 & 10 & 12 & 14 & 16 & 18 & 20 & 22 & 24 & 26 & 28 & 30 & 32 \end{array} \right)$$

Permute columns of the matrix $\mathbf{E}_{32}$ according to permutation $\pi_{32}$. As a result of such permutation, we obtain the matrix

$$\tilde{\mathbf{E}}_{32} = \left[ \begin{array}{c:c} \mathbf{E}_{16} & \mathbf{Q}_{16} \\ \hdashline \mathbf{E}_{16} & -\mathbf{Q}_{16} \end{array} \right] = \mathbf{E}_{32}\mathbf{P}_{32}^{(\pi_{32})}$$

According to the concept, expression (4) for a given transform size can be rewritten as follows:

$$\mathbf{Y}_{32\times 1} = \mathbf{W}_{32}^{(0)}(\mathbf{E}_{16} \oplus \mathbf{Q}_{16})\mathbf{P}_{32}^{(\pi_{32})}\mathbf{X}_{32\times 1} \tag{20}$$

where

$$\mathbf{W}_{32}^{(0)} = (\mathbf{H}_2 \otimes \mathbf{I}_{16}),$$

$$\mathbf{P}_{32}^{(\pi_{32})} = \left[ \begin{array}{c} \check{\mathbf{P}}_{16\times 32} \\ \hdashline \hat{\mathbf{P}}_{16\times 32} \end{array} \right], \quad \check{\mathbf{P}}_{16\times 32} = \overset{3}{\underset{i=0}{\oplus}} \check{\mathbf{P}}_{4\times 8}, \quad \hat{\mathbf{P}}_{16\times 32} = \overset{3}{\underset{i=0}{\oplus}} \hat{\mathbf{P}}_{4\times 8},$$

$$\check{\mathbf{P}}_{4\times 8}^{(i)} = \left[ \begin{array}{cccc:cccc} 1 & & & & & & & \\ & 1 & & & & & & \\ \hdashline & & & & 1 & & & \\ & & & & & & & 1 \end{array} \right], \quad \hat{\mathbf{P}}_{4\times 8}^{(i)} = \left[ \begin{array}{cccc:cccc} 1 & & & & & & & \\ & & & & 1 & & & \\ \hdashline & & & & & 1 & & \\ & & & & & & & 1 \end{array} \right],$$

$\mathbf{E}_{16}$ is the same as in the algorithm for $N = 16$, so we will skip this part.

$\mathbf{Q}_{16}$ is a new matrix. Permute rows of the matrix $\mathbf{Q}_{16}$ according to permutation $\pi_{16}$. As a result of such a permutation, we obtain the matrix

$$\tilde{\mathbf{Q}}_{16} = \left[ \begin{array}{c:c} \mathbf{A}_8 & \mathbf{A}_8 \\ \hdashline \mathbf{B}_8 & -\mathbf{B}_8 \end{array} \right] = \mathbf{Q}_{16}\mathbf{P}_{16}^{(\pi_{16})}$$

where

$$\mathbf{A}_8 = \left[ \begin{array}{cccc:cccc} a_{32} & a_{32} & a_{32} & a_{32} & a_{32} & a_{32} & a_{32} & a_{32} \\ j_{32} & k_{32} & l_{32} & m_{32} & -j_{32} & -k_{32} & -l_{32} & -m_{32} \\ n_{32} & o_{32} & -n_{32} & -o_{32} & n_{32} & o_{32} & -n_{32} & o_{32} \\ k_{32} & -j_{32} & -m_{32} & l_{32} & -k_{32} & j_{32} & m_{32} & -l_{32} \\ \hdashline p_{32} & -p_{32} & p_{32} & -p_{32} & p_{32} & -p_{32} & p_{32} & -p_{32} \\ l_{32} & -m_{32} & -j_{32} & k_{32} & -l_{32} & m_{32} & j_{32} & -k_{32} \\ o_{32} & n_{32} & -o_{32} & -n_{32} & o_{32} & n_{32} & -o_{32} & -n_{32} \\ m_{32} & l_{32} & k_{32} & j_{32} & -m_{32} & -l_{32} & -k_{32} & -j_{32} \end{array} \right],$$

and

$$\mathbf{B}_8 = \left[ \begin{array}{cccc:cccc} b_{32} & c_{32} & d_{32} & e_{32} & f_{32} & g_{32} & h_{32} & i_{32} \\ c_{32} & f_{32} & i_{32} & -d_{32} & -g_{32} & b_{32} & e_{32} & h_{32} \\ d_{32} & i_{32} & -f_{32} & c_{32} & h_{32} & -e_{32} & b_{32} & g_{32} \\ e_{32} & -d_{32} & c_{32} & -b_{32} & -i_{32} & h_{32} & -g_{32} & f_{32} \\ \hdashline f_{32} & -g_{32} & h_{32} & -i_{32} & -b_{32} & c_{32} & -d_{32} & e_{32} \\ g_{32} & b_{32} & -e_{32} & h_{32} & c_{32} & -f_{32} & i_{32} & d_{32} \\ h_{32} & e_{32} & b_{32} & -g_{32} & -d_{32} & i_{32} & f_{32} & c_{32} \\ i_{32} & h_{32} & g_{32} & f_{32} & e_{32} & d_{32} & c_{32} & b_{32} \end{array} \right],$$

where

$$a_{32} = 1,$$

$$b_{32} = 0.9808 - j0.1951, \qquad c_{32} = 0.8315 - j0.5556,$$

$$d_{32} = 0.5556 - j0.8315, \qquad e_{32} = 0.1951 - j0.9808,$$

$$f_{32} = -0.1951 - j0.9808, \qquad g_{32} = -0.5556 - j0.8315,$$

$$h_{32} = -0.8315 - j0.5556, \qquad i_{32} = -0.9808 - j0.1951,$$

$$j_{32} = 0.9239 - j0.3827, \qquad k_{32} = 0.3827 - j0.9239,$$

$$l_{32} = -0.3827 - j0.9239, \qquad m_{32} = -0.9239 - j0.3827,$$

$$n_{32} = 0.7071 - j0.7071, \qquad o_{32} = -0.7071 - j0.7071,$$

$$p_{32} = -j,$$

Taking into account the above factorization scheme, we can finally write

$$\mathbf{Y}_{32 \times 1} = \mathbf{W}_{32}^{(0)} \tilde{\mathbf{W}}_{32}^{(0)} (\mathbf{E}_8 \oplus \mathbf{Q}_8 \oplus \mathbf{A}_8 \oplus \mathbf{B}_8) \tilde{\mathbf{W}}_{32}^{(1)} \mathbf{P}_{32}^{(\pi_{32})} \mathbf{X}_{32 \times 1} \qquad (21)$$

where

$$\check{\mathbf{P}}_{8 \times 4} = \begin{bmatrix} 1 & & & \\ & 1 & & \\ \hdashline & & 1 & \\ & & & 1 \\ & & & \\ & & & \end{bmatrix}, \quad \hat{\mathbf{P}}_{8 \times 4} = \begin{bmatrix} 1 & & & \\ & 1 & & \\ \hdashline & & & \\ & & 1 & \\ & & & 1 \end{bmatrix},$$

$$\check{\mathbf{P}}_{16 \times 8} = \check{\mathbf{P}}_{8 \times 4} \oplus \check{\mathbf{P}}_{8 \times 4}, \quad \hat{\mathbf{P}}_{16 \times 8} = \hat{\mathbf{P}}_{8 \times 4} \oplus \hat{\mathbf{P}}_{8 \times 4},$$

$$\dot{\mathbf{P}}_{16} = \begin{bmatrix} \check{\mathbf{P}}_{16 \times 8} & \hat{\mathbf{P}}_{16 \times 8} \end{bmatrix},$$

$$\tilde{\mathbf{W}}_{32}^{(0)} = \mathbf{W}_{16}^{(0)} \oplus \dot{\mathbf{P}}_{16}, \qquad \tilde{\mathbf{W}}_{32}^{(1)} = \dot{\mathbf{P}}_{16} \oplus \mathbf{W}_{16}^{(0)}$$

$\mathbf{E}_8$ and $\mathbf{Q}_8$ are the same as in the algorithm for $N = 16$, so we will skip this part.

$\mathbf{A}_8$ and $\mathbf{B}_8$ are a new matrix from the bottom half of the algorithm for $N = 32$. Permute rows of the matrix $\mathbf{A}_8$ according to permutation $\pi_8$. As a result of such a permutation, we obtain the matrix

$$\tilde{\mathbf{A}}_8 = \begin{bmatrix} \mathbf{F}_4 & \mathbf{F}_4 \\ \hdashline \mathbf{G}_4 & -\mathbf{G}_4 \end{bmatrix} = \mathbf{A}_8 \mathbf{P}_8^{(\pi_8)}$$

where

$$\mathbf{F}_4 = \begin{bmatrix} a_{32} & a_{32} & a_{32} & a_{32} \\ n_{32} & o_{32} & -n_{32} & -o_{32} \\ \hdashline p_{32} & -p_{32} & p_{32} & -p_{32} \\ o_{32} & n_{32} & -o_{32} & -n_{32} \end{bmatrix} \quad \text{and} \quad \mathbf{G}_4 = \begin{bmatrix} j_{32} & k_{32} & l_{32} & m_{32} \\ k_{32} & -j_{32} & -m_{32} & l_{32} \\ \hdashline l_{32} & -m_{32} & -j_{32} & k_{32} \\ m_{32} & l_{32} & k_{32} & j_{32} \end{bmatrix}$$

Let us now define the permutation $\pi_8^{(2)}$ in the following form:

$$\pi_8^{(2)} = \begin{pmatrix} 1 & 2 & 3 & 4 & 5 & 6 & 7 & 8 \\ 1 & 2 & 3 & 4 & 8 & 7 & 6 & 5 \end{pmatrix}.$$

Permutation $\pi_8^{(2)}$ can be written as a matrix in this way:

$$\mathbf{P}_8^{(\pi_8^{(2)})} = \begin{bmatrix} 1 & & & & & & & \\ & 1 & & & & & & \\ & & 1 & & & & & \\ & & & 1 & & & & \\ \hline & & & & & 1 & & \\ & & & & & & 1 & \\ & & & & & 1 & & \\ & & & & 1 & & & \end{bmatrix}.$$

Then, we permute rows and columns of the matrix $\mathbf{B}_8$ according to permutation $\pi_8^{(2)}$. As a result of such a permutation, we obtain the matrix

$$\tilde{\mathbf{B}}_8 = \left[ \begin{array}{c|c} \mathbf{J}_4 & \mathbf{K}_4 \\ \hline \mathbf{K}_4 & \mathbf{J}_4 \end{array} \right] = \mathbf{P}_8^{(\pi_8^{(2)})} \mathbf{B}_8 \mathbf{P}_8^{(\pi_8^{(2)})}$$

where

$$\mathbf{J}_4 = \left[ \begin{array}{cc|cc} b_{32} & c_{32} & d_{32} & e_{32} \\ c_{32} & f_{32} & i_{32} & -d_{32} \\ \hline d_{32} & i_{32} & -f_{32} & c_{32} \\ e_{32} & -d_{32} & c_{32} & -b_{32} \end{array} \right] \quad \text{and} \quad \mathbf{K}_4 = \left[ \begin{array}{cc|cc} i_{32} & h_{32} & g_{32} & f_{32} \\ h_{32} & e_{32} & b_{32} & -g_{32} \\ \hline g_{32} & b_{32} & -e_{32} & h_{32} \\ f_{32} & -g_{32} & h_{32} & -i_{32} \end{array} \right]$$

To reduce the computational complexity for matrix $\mathbf{B}_8$, we need to perform the following calculations:

$$\tilde{\mathbf{B}}_8 = \left[ \begin{array}{c|c} \mathbf{J}_4 & \mathbf{K}_4 \\ \hline \mathbf{K}_4 & \mathbf{J}_4 \end{array} \right] = \mathbf{H}_8 \left( \check{\mathbf{J}}_4 \oplus \tilde{\mathbf{K}}_4 \right) \mathbf{H}_8$$

where

$$\check{\mathbf{J}}_4 = \frac{1}{2}(\mathbf{J}_4 + \mathbf{K}_4) = \frac{1}{2} \left[ \begin{array}{cc|cc} b_{32} + i_{32} & c_{32} + h_{32} & d_{32} + g_{32} & e_{32} + f_{32} \\ c_{32} + h_{32} & f_{32} + e_{32} & i_{32} + b_{32} & -d_{32} + -g_{32} \\ \hline d_{32} + g_{32} & i_{32} + b_{32} & -f_{32} + -e_{32} & c_{32} + h_{32} \\ e_{32} + f_{32} & -d_{32} + -g_{32} & c_{32} + h_{32} & -b_{32} + -i_{32} \end{array} \right]$$

and

$$\tilde{\mathbf{K}}_4 = \frac{1}{2}(\mathbf{J}_4 - \mathbf{K}_4) = \frac{1}{2} \left[ \begin{array}{cc|cc} b_{32} - i_{32} & c_{32} - h_{32} & d_{32} - g_{32} & e_{32} - f_{32} \\ c_{32} - h_{32} & f_{32} - e_{32} & i_{32} - b_{32} & -d_{32} + g_{32} \\ \hline d_{32} - g_{32} & i_{32} - b_{32} & -f_{32} + e_{32} & c_{32} - h_{32} \\ e_{32} - f_{32} & -d_{32} + g_{32} & c_{32} - h_{32} & -b_{32} + i_{32} \end{array} \right].$$

Taking into account the above factorization scheme, we can finally write

$$\mathbf{Y}_{32 \times 1} = \mathbf{W}_{32}^{(0)} \tilde{\mathbf{W}}_{32}^{(0)} \mathbf{P}_{32}^{(3)} \mathbf{W}_{32}^{(3)} \mathbf{D}_{32} \mathbf{W}_{32}^{(4)} \mathbf{P}_{32}^{(4)} \tilde{\mathbf{W}}_{32}^{(1)} \mathbf{P}_{32}^{(\pi_{32})} \mathbf{X}_{32 \times 1} \tag{22}$$

where

$$\mathbf{P}_{32}^{(3)} = \mathbf{P}_{16}^{(2)} \oplus \mathbf{P}_8^{(\pi_8)} \oplus \mathbf{P}_8^{(\pi_8^{(2)})}, \qquad \mathbf{W}_{32}^{(3)} = \mathbf{W}_{16}^{(2)} \oplus \mathbf{I}_8 \oplus \mathbf{W}_8^{(0)},$$

$$\mathbf{D}_{32} = \mathbf{A}_4 \oplus \mathbf{B}_4 \oplus \mathbf{C}_4 \oplus \mathbf{D}_4 \oplus \mathbf{F}_4 \oplus \mathbf{G}_4 \oplus \check{\mathbf{J}}_4 \oplus \tilde{\mathbf{K}}_4$$

$$\mathbf{W}_{32}^{(4)} = \mathbf{W}_{16}^{(1)} \oplus \mathbf{W}_8^{(0)} \oplus \mathbf{W}_8^{(0)}, \qquad \mathbf{P}_{32}^{(4)} = \mathbf{P}_{16}^{(1)} \oplus \mathbf{I}_8 \oplus \mathbf{P}_8^{(2)}.$$

where

$$\mathbf{P}_{16}^{(1)} = \mathbf{P}_8^{(\pi_8)} \oplus \mathbf{I}_8, \qquad \mathbf{P}_{16}^{(2)} = \mathbf{I}_8 \oplus \left[ \mathbf{P}_8^{(\pi_8)} \right]^{\mathrm{T}},$$

$$\mathbf{W}_{16}^{(1)} = \mathbf{I}_8 \oplus \mathbf{W}_8^{(0)}, \qquad \mathbf{W}_{16}^{(2)} = \mathbf{W}_8^{(0)} \oplus \mathbf{I}_8,$$

$\mathbf{A}_4, \mathbf{B}_4, \mathbf{C}_4$ and $\mathbf{D}_4$ are the same as in the algorithm for $N = 16$, so we will skip this part. $\mathbf{F}_4, \mathbf{G}_4, \tilde{\mathbf{J}}_4$ and $\tilde{\mathbf{K}}_4$ are a new matrix from the bottom half of the algorithm for $N = 32$. Permute rows of the matrix $\mathbf{F}_4$ according to permutation $\pi_4$. As a result of such a permutation, we obtain the matrix

$$\tilde{\mathbf{F}}_4 = \left[ \begin{array}{c:c} \mathbf{L}_2 & \mathbf{L}_2 \\ \hdashline \mathbf{M}_2 & -\mathbf{M}_2 \end{array} \right] = \mathbf{F}_4 \mathbf{P}_4^{(\pi_4)}$$

where

$$\mathbf{L}_2 = \left[ \begin{array}{c:c} a_{32} & a_{32} \\ \hdashline p_{32} & -p_{32} \end{array} \right] \quad \text{and} \quad \mathbf{M}_2 = \left[ \begin{array}{c:c} n_{32} & o_{32} \\ \hdashline o_{32} & n_{32} \end{array} \right].$$

Permute rows and columns of the matrix $\mathbf{G}_4$ according to permutation $\tilde{\pi}_4$. As a result of such a permutation, we obtain the matrix

$$\tilde{\mathbf{G}}_4 = \left[ \begin{array}{c:c} \mathbf{N}_2 & \mathbf{O}_2 \\ \hdashline \mathbf{O}_2 & \mathbf{N}_2 \end{array} \right] = \mathbf{P}_4^{(\tilde{\pi}_4)} \mathbf{G}_4 \mathbf{P}_4^{(\tilde{\pi}_4)}$$

where

$$\mathbf{N}_2 = \left[ \begin{array}{c:c} j_{32} & k_{32} \\ \hdashline k_{32} & -j_{32} \end{array} \right] \quad \text{and} \quad \mathbf{O}_2 = \left[ \begin{array}{c:c} m_{32} & l_{32} \\ \hdashline l_{32} & -m_{32} \end{array} \right].$$

To reduce the computational complexity for matrix $\mathbf{G}_4$, we need to perform the following calculations:

$$\tilde{\mathbf{G}}_4 = \left[ \begin{array}{c:c} \mathbf{N}_2 & \mathbf{O}_2 \\ \hdashline \mathbf{O}_2 & \mathbf{N}_2 \end{array} \right] = \mathbf{H}_4 (\tilde{\mathbf{N}}_2 \oplus \tilde{\mathbf{O}}_2) \mathbf{H}_4$$

where

$$\tilde{\mathbf{N}}_2 = \frac{1}{2}(\mathbf{N}_2 + \mathbf{O}_2) = \frac{1}{2} \left[ \begin{array}{c:c} j_{32} + m_{32} & k_{32} + l_{32} \\ \hdashline k_{32} + l_{32} & -j_{32} - m_{32} \end{array} \right]$$

and

$$\tilde{\mathbf{O}}_2 = \frac{1}{2}(\mathbf{N}_2 - \mathbf{O}_2) = \frac{1}{2} \left[ \begin{array}{c:c} j_{32} - m_{32} & k_{32} - l_{32} \\ \hdashline k_{32} - l_{32} & -j_{32} + m_{32} \end{array} \right]$$

Let us define the permutations $\pi_4^{(1)}$ and $\pi_4^{(2)}$ in the following form:

$$\pi_4^{(1)} = \left( \begin{array}{cccc} 1 & 2 & 3 & 4 \\ 3 & 2 & 1 & 4 \end{array} \right) \quad \text{and} \quad \pi_4^{(2)} = \left( \begin{array}{cccc} 1 & 2 & 3 & 4 \\ 1 & 4 & 3 & 2 \end{array} \right).$$

Permutations $\pi_4^{(1)}$ and $\pi_4^{(2)}$ can be written as matrices in this way:

$$\mathbf{P}_4^{(\pi_4^{(1)})} = \left[ \begin{array}{cc:cc} & & 1 & \\ & 1 & & \\ \hdashline 1 & & & \\ & & & 1 \end{array} \right] \quad \text{and} \quad \mathbf{P}_4^{(\pi_4^{(2)})} = \left[ \begin{array}{cc:cc} 1 & & & \\ & & & 1 \\ \hdashline & & 1 & \\ & 1 & & \end{array} \right].$$

Permute rows and columns of the matrix $\tilde{\mathbf{J}}_4$ according to permutation $\pi_4^{(1)}$ for rows and $\pi_4^{(2)}$ for columns. As a result of such a permutation, we obtain the matrix

$$\dot{\mathbf{J}}_4 = \left[ \begin{array}{c:c} \mathbf{P}_2 & \mathbf{R}_2 \\ \hdashline \mathbf{S}_2 & \mathbf{P}_2 \end{array} \right] = \mathbf{P}_4^{(\tilde{\pi}_4)} \tilde{\mathbf{J}}_4 \mathbf{P}_4^{(\tilde{\pi}_4)}$$

where

$$\mathbf{P}_2 = \left[ \begin{array}{c:c} d_{32} + g_{32} & c_{32} + h_{32} \\ \hdashline c_{32} + h_{32} & -d_{32} + -g_{32} \end{array} \right], \quad \mathbf{R}_2 = \left[ \begin{array}{c:c} -f_{32} + -e_{32} & b_{32} + i_{32} \\ \hdashline b_{32} + i_{32} & e_{32} + f_{32} \end{array} \right],$$

$$\mathbf{S}_2 = \left[ \begin{array}{c:c} b_{32} + i_{32} & e_{32} + f_{32} \\ \hdashline e_{32} + f_{32} & -b_{32} + -i_{32} \end{array} \right].$$

To reduce the computational complexity for matrix $\dot{\mathbf{J}}_4$, we need to perform the following calculations:

$$\tilde{\mathbf{J}}_4 = \left(\mathbf{T}_{2\times 3}^{(1)} \otimes \mathbf{H}_2\right) \begin{bmatrix} \mathbf{S}_2 - \mathbf{P}_2 & & \\ & \mathbf{R}_2 - \mathbf{P}_2 & \\ & & \mathbf{P}_2 \end{bmatrix} (\mathbf{T}_{3\times 2} \otimes \mathbf{H}_2)$$

where

$$\mathbf{T}_{2\times 3}^{(1)} = \begin{bmatrix} 0 & 1 & 1 \\ 1 & 0 & 1 \end{bmatrix}.$$

The same permutations as on matrix $\tilde{\mathbf{J}}_4$ are applied to matrix $\tilde{\mathbf{K}}_4$. As a result of such permutations, we obtain the matrix

$$\dot{\mathbf{K}}_4 = \begin{bmatrix} \mathbf{T}_2 & \mathbf{U}_2 \\ \mathbf{W}_2 & \mathbf{T}_2 \end{bmatrix} = \mathbf{P}_4^{(\tilde{\pi}_4)} \tilde{\mathbf{K}}_4 \mathbf{P}_4^{(\tilde{\pi}_4)}$$

where

$$\mathbf{T}_2 = \begin{bmatrix} d_{32} - g_{32} & c_{32} - h_{32} \\ c_{32} - h_{32} & -d_{32} + g_{32} \end{bmatrix}, \quad \mathbf{U}_2 = \begin{bmatrix} e_{32} - f_{32} & i_{32} - b_{32} \\ i_{32} - b_{32} & f_{32} - e_{32} \end{bmatrix},$$

$$\mathbf{W}_2 = \begin{bmatrix} b_{32} - i_{32} & e_{32} - f_{32} \\ e_{32} - f_{32} & i_{32} - b_{32} \end{bmatrix}.$$

To reduce the computational complexity for matrix $\dot{\mathbf{K}}_4$, we need to perform the following calculations:

$$\tilde{\mathbf{K}}_4 = \left(\mathbf{T}_{2\times 3}^{(1)} \otimes \mathbf{H}_2\right) \begin{bmatrix} \mathbf{W}_2 - \mathbf{T}_2 & & \\ & \mathbf{U}_2 - \mathbf{T}_2 & \\ & & \mathbf{T}_2 \end{bmatrix} (\mathbf{T}_{3\times 2} \otimes \mathbf{H}_2).$$

Taking into account the above factorization scheme, we can finally write

$$\mathbf{Y}_{32\times 1} = \mathbf{W}_{32}^{(0)} \tilde{\mathbf{W}}_{32}^{(0)} \mathbf{P}_{32}^{(3)} \mathbf{W}_{32}^{(3)} \mathbf{P}_{32}^{(5)} \mathbf{W}_{32\times 36} \mathbf{D}_{36} \mathbf{W}_{36\times 32} \mathbf{P}_{32}^{(6)} \mathbf{W}_{32}^{(4)} \mathbf{P}_{32}^{(4)} \tilde{\mathbf{W}}_{32}^{(1)} \mathbf{P}_{32}^{(\pi_{32})} \mathbf{X}_{32\times 1} \tag{23}$$

where

$$\mathbf{P}_{32}^{(5)} = \mathbf{P}_{16}^{(4)} \oplus \mathbf{P}_4^{(\pi_4)} \oplus \mathbf{P}_4^{(\tilde{\pi}_4)} \oplus \mathbf{P}_4^{(\pi_4^{(1)})} \oplus \mathbf{P}_4^{(\pi_4^{(1)})},$$

$$\mathbf{W}_{32\times 36} = \mathbf{W}_{16}^{(4)} \oplus \mathbf{I}_4 \oplus \mathbf{W}_4^{(0)} \oplus \left(\mathbf{T}_{2\times 3}^{(1)} \otimes \mathbf{H}_2\right) \oplus \left(\mathbf{T}_{2\times 3}^{(1)} \otimes \mathbf{H}_2\right),$$

$$\mathbf{D}_{36} = \mathbf{D}_{18}^{(1)} \oplus \mathbf{D}_{18}^{(2)},$$

$$\mathbf{D}_{18}^{(1)} = \mathbf{A}_2 \oplus \mathbf{B}_2 \oplus \mathbf{C}_2 \oplus \mathbf{D}_2 \oplus \mathbf{F}_2 \oplus \mathbf{G}_2 \oplus \mathbf{J}_2 \oplus \mathbf{K}_2,$$

$$\mathbf{D}_{18}^{(2)} = \mathbf{L}_2 \oplus \mathbf{M}_2 \oplus \mathbf{N}_2 \oplus \mathbf{O}_2 \oplus \mathbf{P}_2 \oplus \mathbf{R}_2 \oplus \mathbf{S}_2 \oplus \mathbf{T}_2 \oplus \mathbf{U}_2 \oplus \mathbf{W}_2,$$

$$\mathbf{W}_{36\times 32} = \mathbf{W}_{16}^{(3)} \oplus \mathbf{W}_4^{(0)} \oplus \mathbf{W}_4^{(0)} \oplus (\mathbf{T}_{3\times 2} \otimes \mathbf{H}_2) \oplus (\mathbf{T}_{3\times 2} \otimes \mathbf{H}_2),$$

$$\mathbf{P}_{32}^{(6)} = \mathbf{P}_{16}^{(3)} \oplus \mathbf{I}_4 \oplus \mathbf{P}_4^{(\tilde{\pi}_4)} \oplus \mathbf{P}_4^{(\pi_4^{(2)})} \oplus \mathbf{P}_4^{(\pi_4^{(2)})}.$$

In turn, the matrices $\mathbf{A}_2, \mathbf{B}_2, \mathbf{C}_2, \mathbf{D}_2, \mathbf{F}_2, \mathbf{G}_2, \mathbf{J}_2$ and $\mathbf{K}_2$ are the same as in the algorithm for $N = 16$, so we will skip this part. The matrices $\mathbf{L}_2, \mathbf{M}_2, \mathbf{N}_2, \mathbf{O}_2, \mathbf{P}_2, \mathbf{R}_2, \mathbf{S}_2, \mathbf{T}_2, \mathbf{U}_2$ and $\mathbf{W}_2$ also have structures that provide effective factorization, which leads to a decrease in the multiplicative complexity of calculations:

$$\mathbf{L}_2 = \begin{bmatrix} a_{32} & a_{32} \\ p_{32} & -p_{32} \end{bmatrix} = (a_{32} \oplus p_{32})\mathbf{H}_2,$$

$$\mathbf{M}_2 = \left[\begin{array}{c|c} n_{32} & o_{32} \\ \hline o_{32} & n_{32} \end{array}\right] = \mathbf{H}_2 \frac{1}{2}[(n_{32} + o_{32}) \oplus (n_{32} - o_{32})]\mathbf{H}_2,$$

$$\mathbf{N}_2 = \frac{1}{2}\left[\begin{array}{c|c} j_{32} + m_{32} & k_{32} + l_{32} \\ \hline k_{32} + l_{32} & -j_{32} - m_{32} \end{array}\right] = \left[\begin{array}{c|c} -j0.3827 & -j0.9239 \\ \hline -j0.9239 & j0.3827 \end{array}\right] = \left[\begin{array}{c|c} -n_{32}^{(0)} & n_{32}^{(1)} \\ \hline n_{32}^{(1)} & n_{32}^{(0)} \end{array}\right] =$$
$$= \mathbf{T}_{2\times 3}\left[\left(-n_{32}^{(0)} - n_{32}^{(1)}\right) \oplus \left[-\left(-n_{32}^{(0)} + n_{32}^{(1)}\right)\right] \oplus n_{32}^{(1)}\right]\mathbf{T}_{3\times 2}$$

$$\mathbf{O}_2 = \frac{1}{2}\left[\begin{array}{c|c} j_{32} - m_{32} & k_{32} - l_{32} \\ \hline k_{32} - l_{32} & -j_{32} + m_{32} \end{array}\right] = \left[\begin{array}{c|c} 0.9239 & 0.3827 \\ \hline 0.3827 & -0.9239 \end{array}\right] = \left[\begin{array}{c|c} o_{32}^{(0)} & o_{32}^{(1)} \\ \hline o_{32}^{(1)} & -o_{32}^{(0)} \end{array}\right] =$$
$$= \mathbf{T}_{2\times 3}\left[\left(o_{32}^{(0)} - o_{32}^{(1)}\right) \oplus \left[-\left(o_{32}^{(0)} + o_{32}^{(1)}\right)\right] \oplus o_{32}^{(1)}\right]\mathbf{T}_{3\times 2}$$

$$\mathbf{P}_2 = \frac{1}{2}\left[\begin{array}{c|c} b_{32} + i_{32} - (d_{32} + g_{32}) & e_{32} + f_{32} - (c_{32} + h_{32}) \\ \hline e_{32} + f_{32} - (c_{32} + h_{32}) & -b_{32} - i_{32} + (d_{32} + g_{32}) \end{array}\right] = \left[\begin{array}{c|c} j0.6364 & j0.4252 \\ \hline j0.4252 & -j0.6364 \end{array}\right] =$$
$$= \left[\begin{array}{c|c} p_{32}^{(0)} & p_{32}^{(1)} \\ \hline p_{32}^{(1)} & -p_{32}^{(0)} \end{array}\right] = \mathbf{T}_{2\times 3}\left[\left(p_{32}^{(0)} - p_{32}^{(1)}\right) \oplus \left[-\left(p_{32}^{(0)} + p_{32}^{(1)}\right)\right] \oplus p_{32}^{(1)}\right]\mathbf{T}_{3\times 2}$$

$$\mathbf{R}_2 = \frac{1}{2}\left[\begin{array}{c|c} -f_{32} - e_{32} - (d_{32} + g_{32}) & b_{32} + i_{32} - (c_{32} + h_{32}) \\ \hline b_{32} + i_{32} - (c_{32} + h_{32}) & f_{32} + e_{32} + (d_{32} + g_{32})) \end{array}\right] = \left[\begin{array}{c|c} j1.8123 & j0.3605 \\ \hline j0.3605 & -j1.8123 \end{array}\right] =$$
$$= \left[\begin{array}{c|c} r_{32}^{(0)} & r_{32}^{(1)} \\ \hline r_{32}^{(1)} & -r_{32}^{(0)} \end{array}\right] = \mathbf{T}_{2\times 3}\left[\left(r_{32}^{(0)} - r_{32}^{(1)}\right) \oplus \left[-\left(r_{32}^{(0)} + r_{32}^{(1)}\right)\right] \oplus r_{32}^{(1)}\right]\mathbf{T}_{3\times 2}$$

$$\mathbf{S}_2 = \frac{1}{2}\left[\begin{array}{c|c} d_{32} + g_{32} & c_{32} + h_{32} \\ \hline c_{32} + h_{32} & -d_{32} - g_{32} \end{array}\right] = \left[\begin{array}{c|c} j0.8315 & -j0.5556 \\ \hline -j0.5556 & -j0.8315 \end{array}\right] = \left[\begin{array}{c|c} s_{32}^{(0)} & s_{32}^{(1)} \\ \hline s_{32}^{(1)} & -s_{32}^{(0)} \end{array}\right] =$$
$$= \mathbf{T}_{2\times 3}\left[\left(s_{32}^{(0)} - s_{32}^{(1)}\right) \oplus \left[-\left(s_{32}^{(0)} + s_{32}^{(1)}\right)\right] \oplus s_{32}^{(1)}\right]\mathbf{T}_{3\times 2}$$

$$\mathbf{T}_2 = \frac{1}{2}\left[\begin{array}{c|c} b_{32} - i_{32} - (d_{32} - g_{32}) & e_{32} - f_{32} - (c_{32} - h_{32}) \\ \hline e_{32} - f_{32} - (c_{32} - h_{32}) & -b_{32} + i_{32} + (d_{32} - g_{32}) \end{array}\right] = \left[\begin{array}{c|c} 0.4252 & -0.6364 \\ \hline -0.6364 & -0.4252 \end{array}\right] =$$
$$= \left[\begin{array}{c|c} t_{32}^{(0)} & t_{32}^{(1)} \\ \hline t_{32}^{(1)} & -t_{32}^{(0)} \end{array}\right] = \mathbf{T}_{2\times 3}\left[\left(t_{32}^{(0)} - t_{32}^{(1)}\right) \oplus \left[-\left(t_{32}^{(0)} + t_{32}^{(1)}\right)\right] \oplus t_{32}^{(1)}\right]\mathbf{T}_{3\times 2}$$

$$\mathbf{U}_2 = \frac{1}{2}\left[\begin{array}{c|c} e_{32} - f_{32} - (d_{32} - g_{32}) & i_{32} - b_{32} - (c_{32} - h_{32}) \\ \hline i_{32} - b_{32} - (c_{32} - h_{32}) & -e_{32} + f_{32} + (d_{32} - g_{32}) \end{array}\right] = \left[\begin{array}{c|c} -0.3605 & -1.8123 \\ \hline -1.8123 & 0.3605 \end{array}\right] =$$
$$= \left[\begin{array}{c|c} u_{32}^{(0)} & u_{32}^{(1)} \\ \hline u_{32}^{(1)} & -u_{32}^{(0)} \end{array}\right] = \mathbf{T}_{2\times 3}\left[\left(u_{32}^{(0)} - u_{32}^{(1)}\right) \oplus \left[-\left(u_{32}^{(0)} + u_{32}^{(1)}\right)\right] \oplus u_{32}^{(1)}\right]\mathbf{T}_{3\times 2}$$

$$\mathbf{W}_2 = \frac{1}{2}\left[\begin{array}{c|c} d_{32} - g_{32} & c_{32} - h_{32} \\ \hline c_{32} - h_{32} & -d_{32} + g_{32} \end{array}\right] = \left[\begin{array}{c|c} 0.5556 & 0.8315 \\ \hline 0.8315 & -0.5556 \end{array}\right] = \left[\begin{array}{c|c} w_{32}^{(0)} & w_{32}^{(1)} \\ \hline w_{32}^{(1)} & -w_{32}^{(0)} \end{array}\right] =$$
$$= \mathbf{T}_{2\times 3}\left[\left(w_{32}^{(0)} - w_{32}^{(1)}\right) \oplus \left[-\left(w_{32}^{(0)} + w_{32}^{(1)}\right)\right] \oplus w_{32}^{(1)}\right]\mathbf{T}_{3\times 2}.$$

Taking into account the above factorization scheme, we can finally write

$$\mathbf{Y}_{32\times 1} = \mathbf{W}_{32}^{(0)}\tilde{\mathbf{W}}_{32}^{(0)}\mathbf{P}_{32}^{(3)}\mathbf{W}_{32}^{(3)}\mathbf{P}_{32}^{(5)}\mathbf{W}_{32\times 36}\mathbf{W}_{36}^{(0)}\mathbf{P}_{36\times 46}\mathbf{D}_{46}\times$$
$$\times\mathbf{P}_{46\times 36}\mathbf{W}_{36}^{(1)}\mathbf{W}_{36\times 32}\mathbf{P}_{32}^{(6)}\mathbf{W}_{32}^{(4)}\mathbf{P}_{32}^{(4)}\tilde{\mathbf{W}}_{32}^{(1)}\mathbf{P}_{32}^{(\pi_{32})}\mathbf{X}_{32\times 1} \tag{24}$$

where

$$\mathbf{W}_{36}^{(0)} = \mathbf{W}_{16}^{(6)} \oplus \mathbf{I}_2 \oplus \mathbf{H}_2 \oplus \mathbf{I}_{16}, \qquad \mathbf{P}_{36\times 46} = \mathbf{A}_{16\times 18} \oplus \mathbf{I}_4 \oplus \left(\mathbf{T}_{2\times 3}^{(4)} \otimes \mathbf{I}_8\right),$$

$$\mathbf{W}_{36}^{(1)} = \mathbf{W}_{16}^{(5)} \oplus \mathbf{H}_2 \oplus \mathbf{H}_2 \oplus \mathbf{I}_{16}, \qquad \mathbf{P}_{46 \times 36} = \mathbf{A}_{18 \times 16} \oplus \mathbf{I}_4 \oplus \left( \mathbf{T}_{3 \times 2}^{(3)} \otimes \mathbf{I}_8 \right),$$

and finally

$$\mathbf{D}_{46} = diag(\varphi_0, \varphi_1, \ldots, \varphi_{45})$$

where

$$\varphi_{18} = a_{32} = 1, \qquad \varphi_{19} = p_{32} = -j, \qquad \varphi_{20} = \frac{1}{2}(n_{32} + o_{32}) = -j0.7071,$$

$$\varphi_{21} = \frac{1}{2}(n_{32} - o_{32}) = 0.7071, \qquad \varphi_{22} = \frac{1}{2}(j_{32} + m_{32} - k_{32} - l_{32}) = j0.5412,$$

$$\varphi_{23} = \frac{1}{2}(-j_{32} - m_{32} - k_{32} - l_{32}) = j1.3066, \qquad \varphi_{24} = \frac{1}{2}(k_{32} + l_{32}) = j0.9239,$$

$$\varphi_{25} = \frac{1}{2}(j_{32} - m_{32} - k_{32} + l_{32}) = 0.5412, \qquad \varphi_{26} = \frac{1}{2}(-j_{32} + m_{32} - k_{32} + l_{32}) = -1.3066,$$

$$\varphi_{27} = \frac{1}{2}(k_{32} - l_{32}) = 0.3827,$$

$$\varphi_{28} = \frac{1}{2}(b_{32} + i_{32} - (d_{32} + g_{32}) - (e_{32} + f_{32} - (c_{32} + h_{32}))) = j1.0616,$$

$$\varphi_{29} = \frac{1}{2}(-(b_{32} + i_{32} - (d_{32} + g_{32}) + (e_{32} + f_{32} - (c_{32} + h_{32})))) = -j0.2112,$$

$$\varphi_{30} = \frac{1}{2}(e_{32} + f_{32} - (c_{32} + h_{32})) = -j0.4252,$$

$$\varphi_{31} = \frac{1}{2}(-f_{32} - e_{32} - (d_{32} + g_{32}) - (b_{32} + i_{32} - (c_{32} + h_{32}))) = j1.4518,$$

$$\varphi_{32} = \frac{1}{2}(-(-f_{32} - e_{32} - (d_{32} + g_{32}) + (b_{32} + i_{32} - (c_{32} + h_{32})))) = -j2.1727,$$

$$\varphi_{33} = \frac{1}{2}(b_{32} + i_{32} - (c_{32} + h_{32})) = j0.3605,$$

$$\varphi_{34} = \frac{1}{2}(d_{32} + g_{32} - (c_{32} + h_{32})) = -j0.2759,$$

$$\varphi_{35} = \frac{1}{2}(-(d_{32} + g_{32} + c_{32} + h_{32})) = j1.3870, \qquad \varphi_{36} = \frac{1}{2}(c_{32} + h_{32}) = -j0.5556,$$

$$\varphi_{37} = \frac{1}{2}(b_{32} - i_{32} - (d_{32} - g_{32}) - (e_{32} - f_{32} - (c_{32} - h_{32}))) = 1.0616,$$

$$\varphi_{38} = \frac{1}{2}(-(b_{32} - i_{32} - (d_{32} - g_{32}) + (e_{32} - f_{32} - (c_{32} - h_{32})))) = 0.2112,$$

$$\varphi_{39} = \frac{1}{2}(e_{32} - f_{32} - (c_{32} - h_{32})) = -0.6364,$$

$$\varphi_{40} = \frac{1}{2}(e_{32} - f_{32} - (d_{32} - g_{32}) - (i_{32} - b_{32} - (c_{32} - h_{32}))) = 1.4518,$$

$$\varphi_{41} = \frac{1}{2}(-(e_{32} - f_{32} - (d_{32} - g_{32}) + (i_{32} - b_{32} - (c_{32} - h_{32})))) = 2.1727,$$

$$\varphi_{42} = \frac{1}{2}(i_{32} - b_{32} - (c_{32} - h_{32})) = -1.8123, \quad \varphi_{43} = \frac{1}{2}(d_{32} - g_{32} - (c_{32} - h_{32})) = -0.2759,$$

$$\varphi_{44} = \frac{1}{2}(-(d_{32} - g_{32} + (c_{32} - h_{32}))) = -1.3870, \qquad \varphi_{45} = \frac{1}{2}(c_{32} - h_{32}) = 0.8315.$$

Figure 4 shows a data flow graph of a synthesized algorithm for the 32-point DFT. As can be seen, in this case, the algorithm takes 36 multiplications and 244 additions.

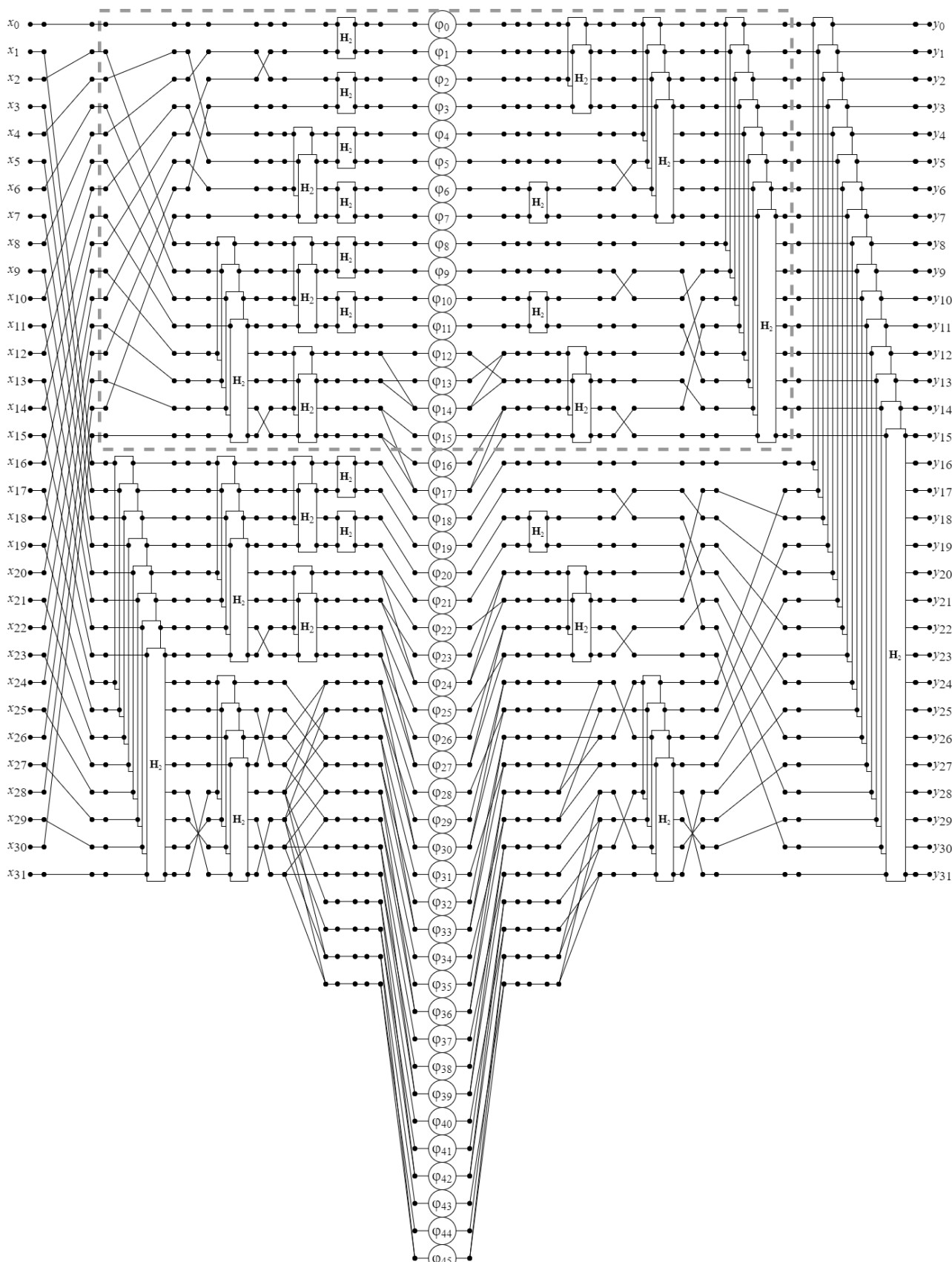

**Figure 4.** The data flow graph of the proposed algorithm for computation of 32-point DFT.

## 5. Conclusions

In this article, we show for the first time a simple, clear and unified approach to the derivation of fast Winograd-like DFT algorithms. The idea of constructing algorithms is based on the application of the method of synthesis of fast algorithms for calculating matrix–vector products described in [19]. The mathematical background for the construction of the described algorithms is the original method of hierarchical factorization of the DFT matrix, which differs from the factorization of this matrix in the case of the Cooley–Tukey FFT. The method of synthesis of algorithms is shown by the examples of the construction of these algorithms for two typical lengths of the initial data sequences: $N = 4$, $N = 8$, $N = 16$ and $N = 32$. As follows from Figure 1, the upper part of the data flow graph for $N = 4$, outlined by the dotted line, corresponds to the algorithm for $N = 2$. In turn, the upper part of the data flow graph for $N = 8$ (see Figure 2), circled with a dotted line, corresponds to the algorithm for $N = 4$. The upper part of the data flow graph for $N = 16$ (Figure 3), circled by a dotted line, corresponds to the algorithm for $N = 8$. Finally, the upper part of the data flow graph for $N = 32$ (Figure 4), circled with a dotted line, corresponds to the algorithm for $N = 16$. It is easy to verify that algorithms for other lengths of sequences that are powers of two can be synthesized in a similar way. Therefore, the described method can be considered as universal.

The advantage of the presented algorithms in comparison with the Cooley–Tukey algorithms is that the critical path in the graph of any of the obtained algorithms contains only one multiplication. If there is more than one multiplication in the critical path of the algorithm, then this will create additional problems for the implementation of computations. As a result of multiplying two $n$-bit operands, a $2n$-bit product is obtained. The need for repeated multiplication requires an additional amount of manipulations with the operands and therefore requires more time and effort than when we are dealing with only a single multiplication. In fixed-point devices, this fact can cause overflow–underflow handling. If we want to preserve the accuracy, then double access to the memory is required both when writing and when reading. Using floating-point arithmetic in this case also creates additional problems related to exponent alignment, mantissa addition, etc. This is what we had in mind when we wrote about additional dignity. Another important advantage of these algorithms over the Cooley–Tukey algorithms is that the multiplications here are either purely real or purely imaginary. Multiplying complex numbers requires three multiplications of real numbers, while multiplying a complex number by a real number requires only two real multiplications. It leads to an additional reduction in the multiplicative complexity of computations. These two advantages are typical of all Winograd-type algorithms.

**Author Contributions:** Conceptualization, A.C.; methodology, A.C. and M.R.; validation, A.C. and M.R.; formal analysis, A.C. and M.R.; investigation, M.R.; writing—original draft, A.C. and M.R.; writing—review and editing, M.R.; supervision, A.C. All authors have read and agreed to the published version of the manuscript.

**Funding:** This research received no external funding.

**Institutional Review Board Statement:** Not applicable.

**Informed Consent Statement:** Not applicable.

**Data Availability Statement:** Not applicable.

**Acknowledgments:** We would like to thank Dorota Majorkowska-Mech for advice and guidance on how to improve the manuscript.

**Conflicts of Interest:** The authors declare no conflict of interest.

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
