# Peer review of "On the Derivation of Winograd-Type DFT Algorithms for Input Sequences Whose Length Is a Power of Two"

_electronics, doi:10.3390/electronics11091342_

Round 1
Reviewer 1 Report
Although there is an element of nesting of each previous value of N within the next higher value of N, there is also a considerable growth of complexity in the upper half of the algorithm, i.e. >N/2 and this seems to be due to involvement of the lower order coefficients in the calculation of terms required for the higher order > N/2 half. In other words, the algorithm does not permit the 'divide and conquer' approach and so becomes increasingly complex.
Is there any general way to deduce the extra factorizations required for the higher order part of the transform? It might be helpful to discuss this if there is.
There do not seem to be any mentions of the actual number of additions vs multiplications required or any comparisons with Cooley-Tukey or other algorithms.
Reviewer 2 Report
There are some aspects that are not very clear
1)
authors say: "In the known papers, the cases of the Winograd FFTs for small sequences of odd length are mainly considered. Moreover, the algorithms were presented in the form of algebraic relations or in the form of the DFT matrix factorizations. However, none of the publications known to
us has written how these relations were obtained or on the basis of any considerations, the matrices that make up the corresponding computational procedures were constructed. In this paper, we want to show a simple, understandable and fairly unified approach
to the derivation of the Winograd-type FFT algorithms for the cases N = 8, N = 16 and N = 32"
The utility to compute odd length FFT is clear. Instead, it is not clear the motivation to discuss the power of two cases.
2) In my opinion there is no necessity to develop all these three cases. A generalization of the math could be better both for readers both for the application of the content of this paper for design.
3)The advantage of the architecture is not put in evidence. Authors develop the DFT for N=4,8 and 16 however, the advantages with respect to traditional FFT are not clear. Why for these values of N designers should choose this method and not the traditional architectures? Please provide comparisons for these cases.
4) Comparisons with the literature are not sufficient
Round 2
Reviewer 2 Report
The article has been improved.